# Fast kinetics of multivalent intercalation chemistry enabled by solvated magnesium-ions into self-established metallic layered materials

Zhenyou Li [1,2], Xiaoke Mu [2], Zhirong Zhao-Karger[1,2], Thomas Diemant[3], R. Jürgen Behm [1,3], Christian Kübel [1,2,4] & Maximilian Fichtner [1,2]

Rechargeable magnesium batteries are one of the most promising candidates for next-generation battery technologies. Despite recent significant progress in the development of efficient electrolytes, an on-going challenge for realization of rechargeable magnesium batteries remains to overcome the sluggish kinetics caused by the strong interaction between double charged magnesium-ions and the intercalation host. Herein, we report that a magnesium battery chemistry with fast intercalation kinetics in the layered molybdenum disulfide structures can be enabled by using solvated magnesium-ions ($[Mg(DME)_x]^{2+}$). Our study demonstrates that the high charge density of magnesium-ion may be mitigated through dimethoxyethane solvation, which avoids the sluggish desolvation process at the cathode-electrolyte interfaces and reduces the trapping force of the cathode lattice to the cations, facilitating magnesium-ion diffusion. The concept of using solvation effect could be a general and effective route to tackle the sluggish intercalation kinetics of magnesium-ions, which can potentially be extended to other host structures.

[1] Helmholtz Institute Ulm (HIU), Helmholtzstraße 11, D-89081 Ulm, Germany. [2] Institute of Nanotechnology (INT), Karlsruhe Institute of Technology (KIT), Hermann-von-Helmholtz-Platz 1, D-76344 Eggenstein-Leopoldshafen, Germany. [3] Institute of Surface Chemistry and Catalysis, Ulm University, Albert-Einstein-Allee 47, D-89081 Ulm, Germany. [4] Karlsruhe Nano Micro Facility (KNMF), Karlsruhe Institute of Technology (KIT), Hermann-von-Helmholtz-Platz 1, D-76344 Eggenstein-Leopoldshafen, Germany. These authors contributed equally: Zhenyou Li, Xiaoke Mu. Correspondence and requests for materials should be addressed to Z.L. (email: zhenyou.li@kit.edu) or to Z.Z-K. (email: zhirong.zhao-karger@kit.edu)

As one of the most promising post-lithium energy storage systems, rechargeable magnesium batteries (RMBs) are highly promising because of their high volumetric capacity (3833 mA h cm$^{-3}$), low reduction potential ($-2.37$ V vs SHE), good safety properties, and resource abundance of Mg metal[1–3]. Since Aurbach et al.[4] proposed the first prototype for RMBs in 2000, it quickly attracted widespread attention. However, the development of RMBs did not meet people's expectation during the past two decades and is still far from the success of either lithium ion batteries (LIBs) or sodium ion batteries (SIBs) to date. A major problem of RMBs lies in the strong electrostatic interaction between the divalent Mg$^{2+}$ and the cathode lattice, resulting in a sluggish kinetics during intercalation into the host material[5]. Actually, the charge density of Mg$^{2+}$ is more than doubled than the Li$^+$ (120 vs 52 C mm$^{-3}$). In addition, a high desolvation energy is usually needed at the liquid–solid interface due to the strong solvation of Mg$^{2+}$ in the electrolyte[6]. Therefore, it is challenging to develop both a cathode material, which is capable of faster Mg$^{2+}$ de-/insertion, and an electrolyte, which can be properly coupled to facilitate Mg$^{2+}$ de-/solvation.

Some recent studies have shown that the kinetic issues could be mitigated simply by performing the electrochemical reaction at elevated temperature[7,8]. However, this induces new difficulties for practical applications, since most of our daily used batteries are running at room temperature. In fact, there is still a lack of suitable cathode materials, which show a comparable battery performance as the Chevrel Phase Mo$_6$S$_8$, the most successful intercalation host so far, for RMBs. Nevertheless, a number of studies have been performed to overcome these intrinsic problems and have shown enhanced electrochemical performances. These include introducing a shielding layer[9,10] or using hydrates[11,12] aiming at reducing the interaction between the Mg$^{2+}$ and the host materials. It is important to note that the shielding molecules should be chosen carefully because they could also participate in the cation shuttle, having a risk to reach the anode side which may cause passivation of the Mg anode. Another effective way to circumvent the problem is to establish a system with modified intercalation chemistry rather than simple Mg$^{2+}$ intercalation. Inspired by the success of AlCl$_4^-$ anions intercalated into graphite in aluminium batteries, Yao's group reported a MgCl$^+$ intercalation chemistry in RMBs with fast kinetics[13]. As an alternative, the concept of introducing additional cations (e.g., Li$^+$) which accelerate reaction kinetics by participating in the cathode reaction results in high-performance hybrid batteries[14]. Furthermore, it is possible to engineer defined defects as intercalation sites to enable the electrochemical reactions in known material systems[15]. Besides, high capacity carbonaceous cathodes have been realized by a fast surface reaction benefiting from the high surface area and the high conductivity of the cathode material[16]. Among them, the MgCl$^+$ intercalation chemistry is the most successful strategy so far. However, the corrosion problem arising from Cl$^-$ ions still makes it difficult for practical application. Therefore, it is still challenging to design a magnesium battery chemistry based on divalent ion intercalation possessing high ion mobility as well as fast electron transfer[17].

To achieve this goal, it is also essential to develop proper electrolytes for RMB cathodes, which offer a wide electrochemical window, suitable cations for intercalation, and a stable contact with both electrodes[18]. The first generation of the electrolyte combined Grignard reagents and Lewis acids[4]. Although intensive efforts have been made to optimize the electrochemical properties[19,20], the nucleophilic electrolytes are still so reactive that only a limited voltage window is available. Non-nucleophilic electrolytes have been proposed using less reactive reagents, which offer a wider electrochemical window and improved chemical stability[21,22]. However, both types of electrolyte are complex, leading to changes in the equilibrium ionic species once the electrochemical reactions take place. Moreover, the addition of chlorides leads to severe corrosion problem, resulting in practical issues[23]. Therefore, pursuing a simple salt, which contains only noncorrosive anions and cations is desirable. Some simple salts including Mg(TFSI)$_2$[24], Mg(BH$_4$)$_2$[25], and Mg(CB$_{11}$H$_{12}$)$_2$[26] have already shown some degree of reversible Mg deposition. But they suffer from either passivation layers on the Mg anode or complicated synthetic procedures[27]. Recently, our group reported a simple synthesis of a new class of non-corrosive, fluorinated Mg alkoxyborate (MgBOR) electrolytes with a wide voltage window up to 4.5 V which is a promising electrolyte for both intercalation type RMBs and Mg–S batteries[28]. Interestingly, the dimethoxyethane (DME), which is used as the solvent for the synthesis of MgBOR, bonds tightly with the Mg$^{2+}$ to form [Mg(DME)$_3$]$^{2+}$ cations either in the salt or in the electrolyte.

Based on this electrolyte, we propose an intercalation chemistry in RMBs, in which solvated divalent [Mg(DME)$_3$]$^{2+}$ ions are inserted into the layered material. Compared to the bare magnesium-ions, the solvated ions have a larger ionic radius and thus reduced charge density, resulting in a lower diffusion energy barrier through the host. With the solvated ion insertion, the sluggish desolvation process is effectively avoided. Compared to other approaches reported for solvated ion intercalation systems (e.g., [Mg(H$_2$O)$_x$]$^{2+}$ and MgCl$^+$), this chemistry neither results in a passivation risk nor a corrosion problem, making it a promising solution for the practical application. The concept is verified using a nanostructured MoS$_2$ cathode (MoS$_2$@C porous nanorods, MoS$_2$@C-PNR), which demonstrates a prominent improvement of battery performance including high capacity of 120 mA h g$^{-1}$, long-term cycling stability for up to 200 cycles, and high rate performance at 0.5 A g$^{-1}$. The solvated ions intercalation process has been thoroughly investigated through various spectroscopic, microscopic, and electrochemical studies as well as statistical analysis. In addition, an intercalation induced phase transition of MoS$_2$ from semi-conductive 2H to metallic 1T phase during intercalation of the solvated magnesium-ions has been observed for the first time in a multivalent intercalation system. The clarification of the Mg storage in MoS$_2$@C-PNR bridges the gap in mechanistic studies of MoS$_2$-based cathodes in RMBs. Importantly, these strategies and experiences can be potentially extended to other multivalent ion intercalation systems.

## Results

**Structural characterization of MoS$_2$@C-PNR.** The crystal structures of the as-synthesized MoS$_2$@C-PNR and of commercial MoS$_2$ powder (MoS$_2$-com) were characterized by X-ray powder diffraction (XRD) (Fig. 1a). The prominent peaks in MoS$_2$@C-PNR could be indexed as (100), (103), and (110) planes of 2H-MoS$_2$ (JCPDS # 37-1492), which indicates the formation of the hexagonal layered MoS$_2$. The broad peaks indicate the small size of the coherently scattering domains. According to the Scherrer equation, the crystal size along (100) and (110) direction was determined to be 4.8 and 4.3 nm, respectively. The absence of the (002) reflection can be explained by the few-layer nature of the MoS$_2$ with strong distortion in the composite as shown in the HRTEM (Fig. 1f). In comparison, all the diffraction peaks of the MoS$_2$-com match well with the standard pattern of 2H-MoS$_2$, which confirms the higher crystallinity of the commercial sample compared to the as-synthesized sample. The chemical state of the composite was further analyzed by high-resolution X-ray photoemission spectroscopy (XPS) detail measurements in the Mo 3$d$ and S 2$p$ regions. The Mo 3$d$ spectrum (Fig. 1b) features a dominant doublet at 229.6 eV (Mo 3$d_{5/2}$) and 232.7 eV (Mo 3$d_{3/2}$), which is attributed to MoS$_2$. Trace amounts of MoO$_3$ were also

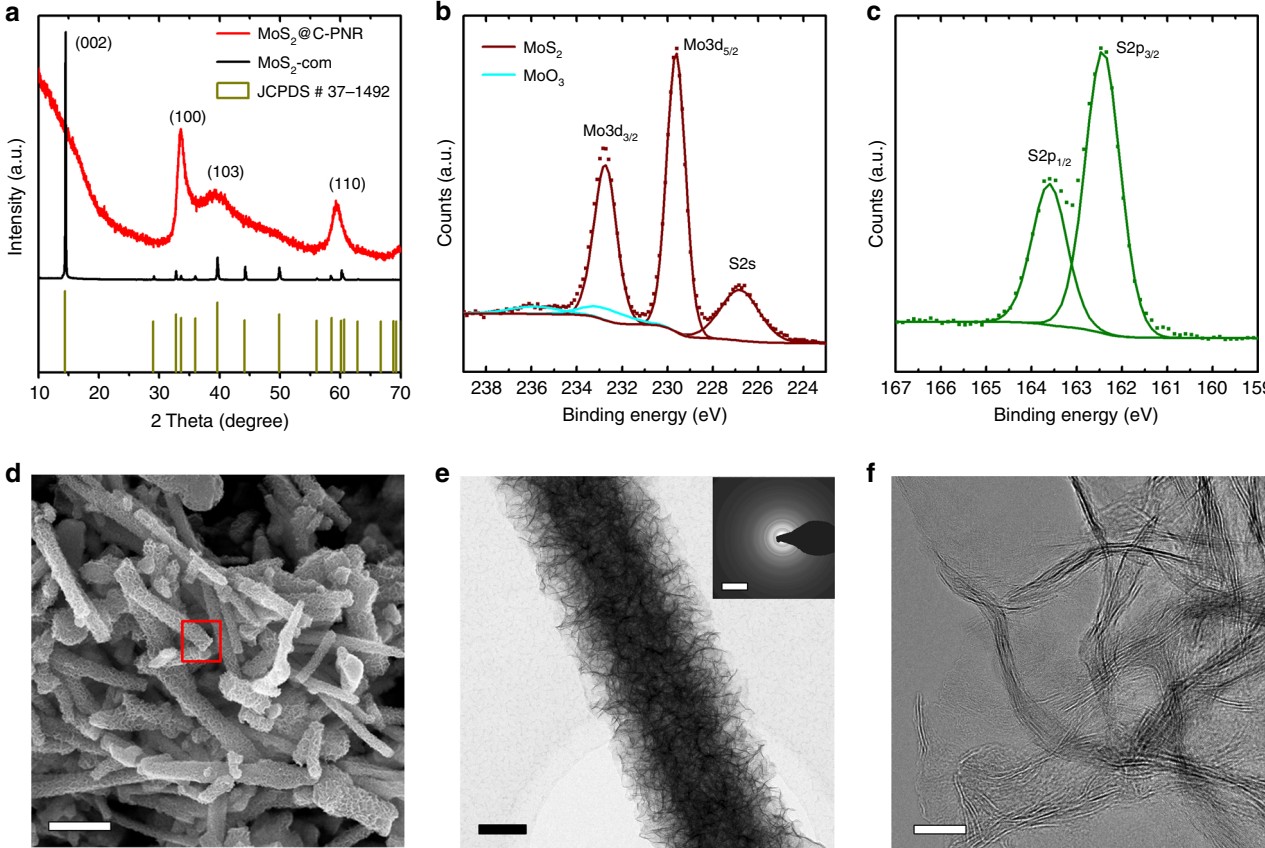

**Fig. 1** Structural characterization of MoS$_2$@C-PNR. **a** XRD patterns (red) and comparison with MoS$_2$-com (black) and standard 2H-MoS$_2$ (dark yellow, JCPDS #37-1492); **b**, **c** XPS Mo 3d and S 2p spectra; **d** representative SEM image (scale bar = 2 μm), the EDX spectrum shown in Supplementary Fig. 2 were recorded in the area of the red square; **e** TEM image (scale bar = 200 nm) with SEAD pattern insert (scale bar = 10 1/nm) and **f** HRTEM image (scale bar = 10 nm)

detected leading to peaks at 232.7 and 235.8 eV. In addition, the small peak at 226.8 eV corresponds to the signal from S 2s. In the S 2p spectrum (Fig. 1c), only the S 2p peak doublet of MoS$_2$ can be detected (S 2p$_{3/2}$ at 162.4 eV and S 2p$_{1/2}$ at 163.6 eV)[29]. The carbon content of the MoS$_2$@C-PNR was determined from the thermogravimetric analysis (TGA) curves shown in Supplementary Fig. 1, in which the two exothermic peaks correspond to the oxidation of carbon and MoS$_2$, respectively. Compared to the MoS$_2$-com sample, the oxidation of MoS$_2$ in MoS$_2$@C-PNR took place at a lower temperature, probably as a result of the nanosized structure. Based on the weight loss, the carbon content in the MoS$_2$@C-PNR sample was calculated to be 17.5% in weight assuming all carbon was burned and all MoS$_2$ was fully oxidized to MoO$_3$.

The morphology of the MoS$_2$@C-PNR was characterized by scanning electron microscopy (SEM) and transmission electron microscopy (TEM). In Fig. 1d, the representative SEM image shows an ordered rod-like structure with several μm in length and 500–800 nm in diameter. The EDX spectrum (Supplementary Fig. 2) of the selected region (red square) indicates Mo and S are dominant in the nanorods with a small amount of carbon and oxygen presented. The MoS$_2$@C-PNR is actually an assembly of the MoS$_2$ nanosheets building blocks. The bright field TEM image (Fig. 1e) reveals an evenly distributed nanosheets array decorated on the surface of the rod, resulting in an open network of MoS$_2$ nanosheets. Selected area electron diffraction (SEAD) demonstrates the polycrystalline nature of MoS$_2$. The HRTEM image in Fig. 1f shows defect-rich nature of the MoS$_2$ sheets. The strongly distorted lattice fringes correspond to the (002) plane of MoS$_2$

and reveal the very limited thickness of the MoS$_2$ sheets along the c-axis, consisting of only a few layers of MoS$_2$. This is consistent with the absence of the (002) peak in XRD. The coherent domain size is less than 10 nm in agreement with the XRD data.

**Electrochemical performance of MoS$_2$@C-PNR.** The electrochemical processes of the MoS$_2$@C-PNR cathode were investigated using a non-corrosive, boron-centered electrolyte (MgBOR in DME)[28]. In the first step, the Mg plating/stripping properties of the pure electrolyte were checked before applying it to the MoS$_2$@C-PNR system. In Supplementary Fig. 3a, the cyclic voltammetry (CV) of stainless steel (SS) against Mg foil with the MgBOR electrolyte shows highly symmetric Mg/Mg$^{2+}$ redox peaks with a large current response of ~30 mA at a scan rate of 25 mV s$^{-1}$, indicating a fast and efficient Mg deposition/dissolution. The high coulombic efficiency of the electrolyte has already been reported in the previous work[28]. Furthermore, the highly reversible plating/stripping process is also evidenced by the tiny amount of net charge accumulation after a complete CV cycle as shown in Supplementary Fig. 3b. The onset of magnesium deposition is −0.4 V from the 2nd cycle on, resulting in an overpotential of ~0.4 V, which is comparable to most of the magnesium battery electrolytes (Supplementary Table 1). During cycling, no other redox peak appears in the whole voltage range, suggesting high oxidation stability up to 4.5 V against SS. Compared to Cl containing electrolyte (APC[4], HMDS[21], etc.), the new electrolyte is non-corrosive and shows remarkably high reversibility and wide electrochemical window, which is desirable for magnesium battery systems.

The three-electrode CV of the MoS$_2$@C-PNR cathode against the reference electrode (Mg$_{RE}$) using 0.4 M MgBOR in DME as electrolyte is presented in Fig. 2a. The reduction peak ($R_1$) at 1.10 V and the oxidation peak ($O_1$) at 1.64 V are presumably related to the insertion and removal of Mg$^{2+}$ through the layered structure[30]. The corresponding CV curve against the counter electrode (Mg$_{CE}$) is shown in Supplementary Fig. 4. Here, the $R_1$ stays at similar position while $O_1$ shifts to 1.85 V. The shift of the oxidation peak is attributed to the overpotential of Mg plating, which was also observed in the CV of Mg stripping/plating (Supplementary Fig. 3a). In both the reduction and the oxidation scan, the potential is ~0.2 V beyond the cut-off potential (0.01–2.5 V vs Mg$_{RE}$/Mg$^{2+}$ as shown in Fig. 2a), which is due to the polarization of the electrolyte[31]. Notably, the redox peaks become more and more prominent upon cycling, indicating an activation process. Another large reduction peak ($R_2$) at around 0.1 V may be the result from a phase transition from 2H-MoS$_2$ to 1T-MoS$_2$ which was previously also observed in LIBs[32,33]. The corresponding oxidation peak ($O_2$) at 0.7 V increases simultaneously with the reduction peak but with less intensity, indicating a partially reversible phase transition back to 2H-MoS$_2$. Figure 2b exhibits typical galvanostatic cycling curves of MoS$_2$@C-PNR at 10 mA g$^{-1}$, in which the charge and discharge plateaus at around 1.1 and 1.8 V match well with the CV data. The initial discharge capacity of the battery is 66 mA h g$^{-1}$, increasing to 118 mA h g$^{-1}$ after 30 cycles, which is associated to an intercalation of 0.36 Mg$^{2+}$ per formula unit. The corresponding cycling performance in Fig. 2c (green dots) also shows an increasing capacity in the beginning which stays stable at ~120 mA h g$^{-1}$ for the first 100 cycles. For comparison, the MoS$_2$-com electrode only has a capacity of 35 mA h g$^{-1}$ after 50 cycles at the

same current rate as shown in Supplementary Fig. 5a. When comparing the charge–discharge profiles of the two electrodes at the 50th cycle (shown in Supplementary Fig. 5b), the main difference is that the MoS$_2$-com electrode exhibits much smaller redox plateaus at 1.1/1.8 V than the MoS$_2$@C-PNR, which are related to the Mg$^{2+}$ intercalation. Therefore, the largely enhanced capacity might be the result of the well-established nanosized structure with a larger specific surface area, which provides a larger contact area with the electrolyte and reduces the Mg diffusion pathway. In order to clarify any contribution of the carbon species in MoS$_2$@C-PNR, the electrochemical performance of a carbon-free MoS$_2$ nanostructure (s-MoS$_2$) and a pure carbon sample (s-C) are shown in Supplementary Fig. 6. The s-MoS$_2$ electrode exhibits similar charge–discharge behavior and comparable capacity (~100 mA h g$^{-1}$ at 20 mA g$^{-1}$ after 50 cycles) as the MoS$_2$@C-PNR electrode. In contrast, the s-C electrode exhibits a much smaller capacity of less than 5 mA h g$^{-1}$. These results prove that the contribution of carbon species to the total capacity of MoS$_2$@C-PNR is negligible.

It should be noted that similar activation behavior was observed when the current rate was increased to 20 mA g$^{-1}$ (brown dots in Fig. 2c): the capacity is increasing within the first 15–20 cycles and stabilizes at ~120 mA h g$^{-1}$ afterwards. The reasons for the activation process will be discussed in detail later in the Mg storage mechanism part. All cells which ran at high current rates (>20 mA g$^{-1}$) were activated at 20 mA g$^{-1}$ for 15 cycles before using the designated current. With this approach, the MoS$_2$@C-PNR electrode running at 50 mA g$^{-1}$ delivers a stable capacity of 85 mA h g$^{-1}$ after activation for up to 100 cycles. The long cycle life is even maintained at higher currents: in Fig. 2d, batteries running at 100, 200, and 500 mA g$^{-1}$ provide capacity retention of 71%

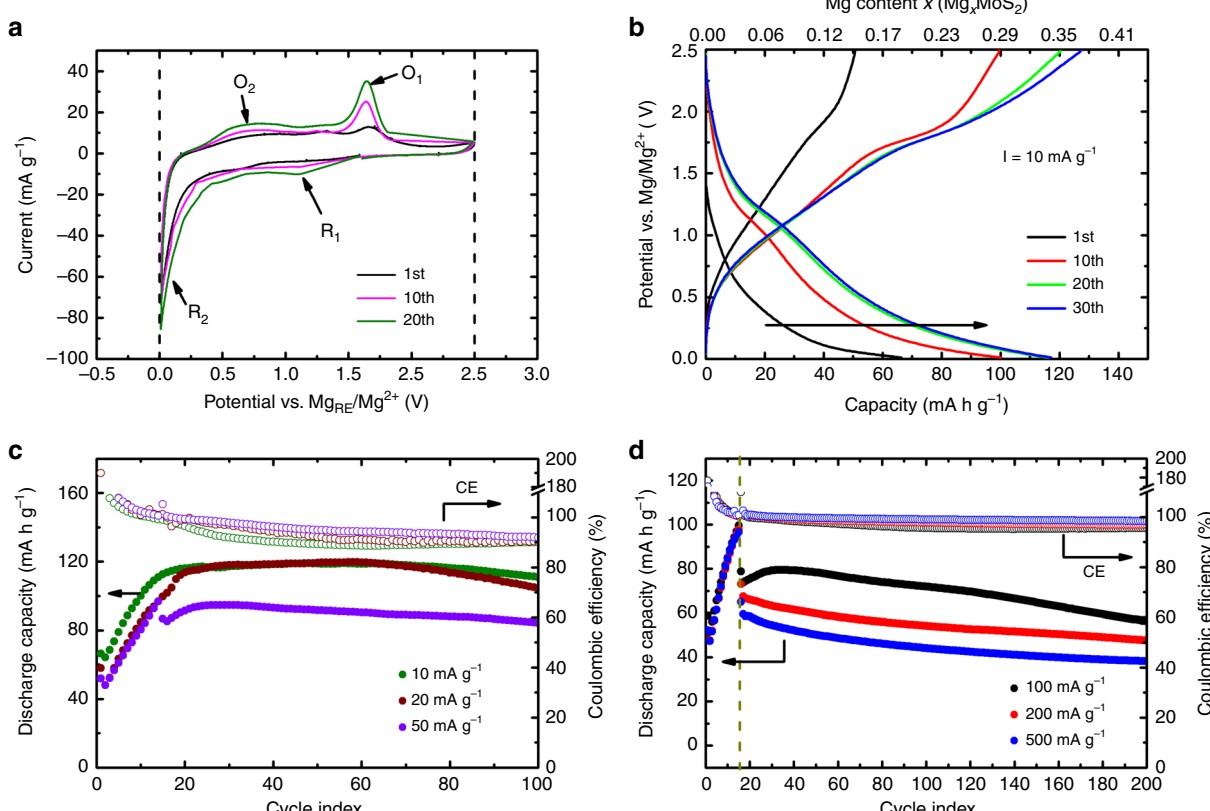

**Fig. 2** Electrochemical performance of MoS$_2$@C-PNR. **a** Three-electrode CV (vs Mg$_{RE}$/Mg$^{2+}$) at a scan rate of 0.1 mV s$^{-1}$ between 0.01 and 2.5 V; **b** selected cycles of galvanostatic cycling at 10 mA g$^{-1}$; **c, d** cycling performance at different current rate, for measurements running at 50, 100, 200 and 500 mA g$^{-1}$, the cells were activated at 20 mA g$^{-1}$ for 15 cycles before applying corresponding current

(56 mA h g$^{-1}$), 65% (48 mA h g$^{-1}$), and 59% (38 mA h g$^{-1}$) after 200 cycles with high coulombic efficiency (CE > 95%). Furthermore, the corresponding energy density and energy efficiency are shown in Supplementary Fig. 7. The cell could deliver a high energy density of almost 80 W h kg$^{-1}$ with an efficiency of ~40% at 20 mA g$^{-1}$. The low energy efficiency is mainly related to the overpotential, which needs to be improved in the follow-up studies. Nevertheless, these promising results make it clear that the battery is capable of fast charge–discharge. In addition, the MoS$_2$@C-PNR electrode also exhibits high rate capability as shown in Supplementary Fig. 8a. With a current density of 10, 20, 50, 100, 200, and 500 mA g$^{-1}$, the battery provides a stable capacity of 130, 115, 91, 74, 60, and 47 mA h g$^{-1}$, respectively. Furthermore, when the current is reverted to 10 mA g$^{-1}$, a reversible capacity of 119 mA h g$^{-1}$ was restored. The charge–discharge profiles at different current rates in Supplementary Fig. 8b show similar behavior. Even at 500 mA g$^{-1}$, the plateaus are still recognizable, which indicates a fast kinetics of the MoS$_2$@C-PNR electrode. The fast kinetics was double checked by CVs with different scan rate (Supplementary Fig. 9), in which the redox peaks persist even at 2 mV s$^{-1}$. In fact, the kinetics could be further enhanced at elevated temperature. As shown in Supplementary Fig. 10, the capacity of the MoS$_2$@C-PNR electrode was largely improved when the temperature was increased to 45 °C: the cell delivered a stable capacity of 160 mA h g$^{-1}$ at 50 mA g$^{-1}$ for up to 50 cycles.

Finally, the high performance of the electrolyte was highlighted by comparing the electrochemical performance of the MoS$_2$@C-PNR with the most commonly used electrolytes (such as APC, HMDS, Mg(TFSI)$_2$). As shown in Supplementary Fig. 11, only with the MgBOR electrolyte, the plateaus are visible in the charge–discharge profile. With the APC electrolyte, the slope behavior results in a low energy density although a comparable capacity was obtained. In the same cell configuration, the MgBOR electrolyte could offer almost double the capacity above 0.5 V compared to APC and was even more improved compared to the other two electrolytes. As a result, the electrode delivered the highest capacity and energy density (60 W h kg$^{-1}$ at 50 mA g$^{-1}$) when coupled with MgBOR electrolyte.

**Fast kinetics of the Mg battery chemistry in MoS$_2$@C-PNR.** In order to clarify the mechanism of the fast kinetics of the Mg battery, different ex situ characterization techniques as well as in situ electrochemical measurements have been performed. Ex situ XRD measurements were taken at the fully discharged or charged states of specific cycles during the activation process. Due to the few-layer nature of the as-synthesized material, the diffraction peaks along the c-axis, which indicate the change of the interlayer distance, are not visible. Therefore, it is not possible to get information about the interlayer expansion/shrinking by XRD. However, it could still be seen that all the other diffraction peaks of MoS$_2$ were fully preserved both at the charged and discharged state during cycling as shown in Supplementary Fig. 12. Moreover, no new peaks corresponding to other species (e.g., S, Mo, or MgS$_x$) were observed except for those at 43.5° and 50.7°, which originate from the SS current collector whose XRD pattern is shown in Supplementary Fig. 13. These results potentially suggest that the MoS$_2$@C-PNR electrode did not undergo a conversion reaction. However, the XRD results only reflect the change of the bulk of the electrode. Details of the surface evolution have been examined further.

In order to confirm this conclusion, ex situ XPS measurements of electrodes at the same dis-/charge states as for the ex situ XRD were performed. Supplementary Fig. 14 presents the Mg 2s spectra of MoS$_2$@C-PNR electrodes at each charge state. The Mg 2s peak at ~89.8 eV becomes prominent upon cycling for the

discharged samples but is negligible at charged states. This proofs the Mg storage in the MoS$_2$@C-PNR cathode. The successive increase of the Mg 2s peak intensity (and the Mg 2s/Mo 3d peak intensity ratio) is consistent with the increasing capacity. The XPS spectra in the Mo 3d and S 2p range are shown in Fig. 3b. As expected, the Mo 3d spectrum of the pristine electrode is identical to the results of the measurements of the MoS$_2$@C-PNR material before electrode preparation and is dominated by the Mo 3d$_{5/2}$ and Mo 3d$_{3/2}$ peak components of 2H-MoS$_2$ at 229.6 and 232.7 eV. In addition, there are again features related to the S 2s orbital (at 226.8 eV) and a trace amount of MoO$_3$ involved, which originate from the unreacted starting material. Upon cycling, the features related to Mo 3d$_{5/2}$ and Mo 3d$_{3/2}$ broaden and shift to lower binding energy, indicating a change of the chemical environment. Deconvolution of the doublet reveals the emergence of an additional doublet at lower binding energy. The new doublet appears at a position ~0.7 eV lower than in 2H-MoS$_2$ and becomes more prominent with cycling. Interestingly, a similar peak shift is observed in the S 2p region. According to the deconvolution of the S 2p spectra of the cycled electrodes, a pair of new peaks (at 162.0 and 163.2 eV) appears in addition to that of 2H-MoS$_2$ (at 162.5 and 163.7 eV). Since the development of the spectra in the Mo 3d and S 2p range is obviously coupled, the same intensity ratio was used in both spectra for a given sample. The simultaneous shift of the peaks in both regions was already seen before for lithiated MoS$_2$ and assigned to the transformation from 2H to 1T phase[34]. It may be noted that, to the best of our knowledge, this is the first time that a 2H to 1T phase transition has been reported in metal sulfide cathodes for a multivalent ion battery system.

The 2H to 1T transition is indicative of electron doping to the 2H-MoS$_2$ during intercalation[33]. As illustrated in Fig. 3c, the 2H-MoS$_2$, which is the thermodynamic stable phase of MoS$_2$, has a trigonal prismatic structure with six neighboring sulfur atoms coordinating each molybdenum atom. The Mo 4d orbitals in 2H-MoS$_2$ are split into three groups, namely (i) $d_{z^2}$, (ii) $d_{xy} + d_{x^2-y^2}$, and (iii) $d_{yz} + d_{xz}$. There is ~1 eV energy gap between the first two band groups. The two Mo d electrons fill the lower $d_{z^2}$ orbital, which makes it semiconductive. For 1T-MoS$_2$, there are only two band groups ($d_{xy} + d_{xz} + d_{yz}$ and $d_{x^2-y^2} + d_{z^2}$) because of its octahedral coordination (Fig. 3d). The two Mo d electrons occupy the $d_{xy} d_{xz} d_{yz}$ orbitals but will not fill them. With the unfilled orbitals, 1T-MoS$_2$ is conductive but thermodynamically not stable. With the Mg$^{2+}$ intercalation, the additional charges from cations will be transferred to MoS$_2$ occupying the $d_{xy} d_{xz} d_{yz}$ orbitals which stabilizes the 1T structure and reduces the valence of Mo atoms[35,36]. The transition from the semiconductive phase to the metallic phase allows for much faster electron transfer, which enhanced the kinetics. It is also important to note that we were not able to detect features in the spectra of the cycled samples which could be assigned to the formation of elemental S (S 2p$_{3/2}$ peak at 164.0 eV) or metallic Mo (Mo 3d$_{5/2}$ peak at 228.0 eV), further confirming that no conversion reaction is taking place[37]. Therefore, the Mg$^{2+}$ intercalation mechanism is evidenced based on the XPS results together with the ever-increasing redox peaks (O$_1$ and R$_1$ in Fig. 2a) in the first few cycles.

The phase transition from 2H to 1T is probably one of the reasons for the large improvement of the electrochemical reaction by increasing the intrinsic conductivity of the cathodes. According to the XPS peak fit, the fraction of 1T-MoS$_2$ within the surface layer probed by XPS increases to almost 70% of the total MoS$_2$ in the measurements of the samples subjected to 30 electrochemical cycles (Fig. 3e). The fraction of 1T-MoS$_2$ seems to depend mainly on the number of electrochemical cycles, while the charge state of the samples within the same electrochemical cycle

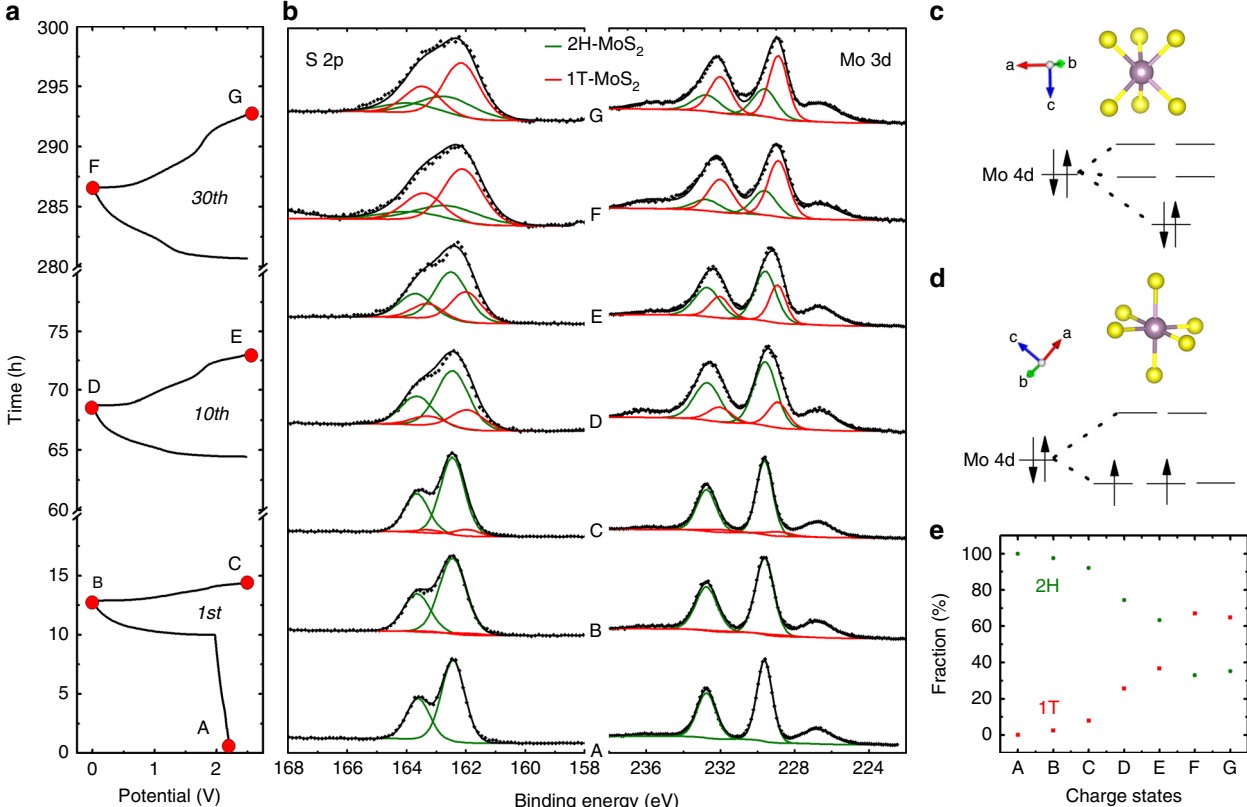

**Fig. 3** Ex situ XPS measurement of MoS$_2$@C-PNR electrodes. **a** Dis-/charge profile of MoS$_2$@C-PNR electrodes at selected cycles. **b** Ex situ XPS spectra of MoS$_2$@C-PNR electrodes at specific charge states indicated in the dis-/charge profile. Structure and Mo 4$d$ orbitals of **c** 2H-MoS$_2$ and **d** 1T-MoS$_2$. **e** The fractions of the two MoS$_2$ components at each charge state

is of minor importance and only a slight increase of the 1T fraction is observed when comparing the discharged and charged samples of the 1st and 10th cycle. The stabilization of the 1T phase during dis-/charge is surprising because the 2H phase is thermodynamically favorable at room temperature. A previous study indicated that 0.2 Li$^+$ (corresponds to 0.1 Mg$^{2+}$) per MoS$_2$ unit is enough to induce the 2H to 1T transition[38]. Therefore, the Mg residues (Mg/Mo ratio) at the surface of the charged states observed by XPS (Supplementary Fig. 14), are enough to stabilize the 1T phase on the surface. As XPS is a surface sensitive technique with limited detection depth <5 nm, the phase evolution taking place below the top surface of the material during dis/charging was examined by S/TEM techniques. The results are discussed in detail in the following part.

To get more insight into the local structure changes during electrochemical cycling, especially in the activated material underneath the top surface, a variety of S/TEM analysis were performed. As shown in the STEM high angle annular dark field (HAADF) images in Fig. 4a–c, the morphology of MoS$_2$@C-PNR changes dramatically from the pristine material to the 30th discharged state. Especially, the nanosheet structure becomes fractured forming a shell zone located from the surface to ~100 nm in depth. The Fast Fourier Transform (FFT) inset of the shell zone in the corresponding HRTEM images exhibits a transition from a clear crystalline state with six-fold symmetry in the pristine material to a poorly defined FFT and finally even amorphous form at the 30th discharged state. These findings indicate a fragmentation and structural distortion process. The EDX spectra of the selected area of the 30th discharged sample (Fig. 4e) and corresponding quantification (Supplementary Fig. 16a) show that the Mg concentration of the fractured areas (areas 2 and 3) is much higher than that in the bulk (area 1).

Previous work has shown that the distortion and amorphization of the MoS$_2$ structure during Na$^+$ insertion are resulting in the 2H to 1T phase transition[39]. Therefore, the fragmentation of the MoS$_2$ nanosheets could be attributed to a Mg$^{2+}$ intercalation-induced 2H to 1T phase transition. The shell zone corresponds to the activated material, in which the electrochemical reaction occurred during cycling. In the STEM-HAADF images in Fig. 4c, yellow dashed lines are used to label the border between the fractured shell and the well-crystallized core where the material was not activated and hence did not react. It clearly shows that the fractured shell becomes thicker upon successive cycling. This change coincides with the capacity increase, explaining the activation process. In contrast, the activated material in the shell of the charged MoS$_2$ sheets becomes more crystalline than that of the discharged counterpart as shown by more distinguishable layered contrast and a more crystalline FFT in Fig. 4d and 4D-STEM patterns in Supplementary Fig. 17. This reveals a MoS$_2$ recrystallization during the charging process. The reformation of the MoS$_2$ sheets in the shell is not complete. Instead, smaller sized MoS$_2$ nanocrystals are formed, which increase the contact with the electrolyte and reduce the necessary Mg$^{2+}$ diffusion distance, further promoting the Mg$^{2+}$ intercalation kinetics. Moreover, the Mg peak in the charged state (Fig. 4f) is almost invisible in contrast to the discharged state (Fig. 4e), which confirms a highly reversible Mg storage.

EDX elemental maps and corresponding STEM-HAADF images of MoS$_2$@C-PNR electrodes at the 30th and 10th discharged state are shown in Fig. 5a, c. The elemental maps indicate that Mo and S are distributed all through the rod-like structure, while Mg$^{2+}$ is mainly concentrated at the shell zone with thickness ~100 nm. All the results indicate a reaction gradually progressing from the surface to the bulk in agreement

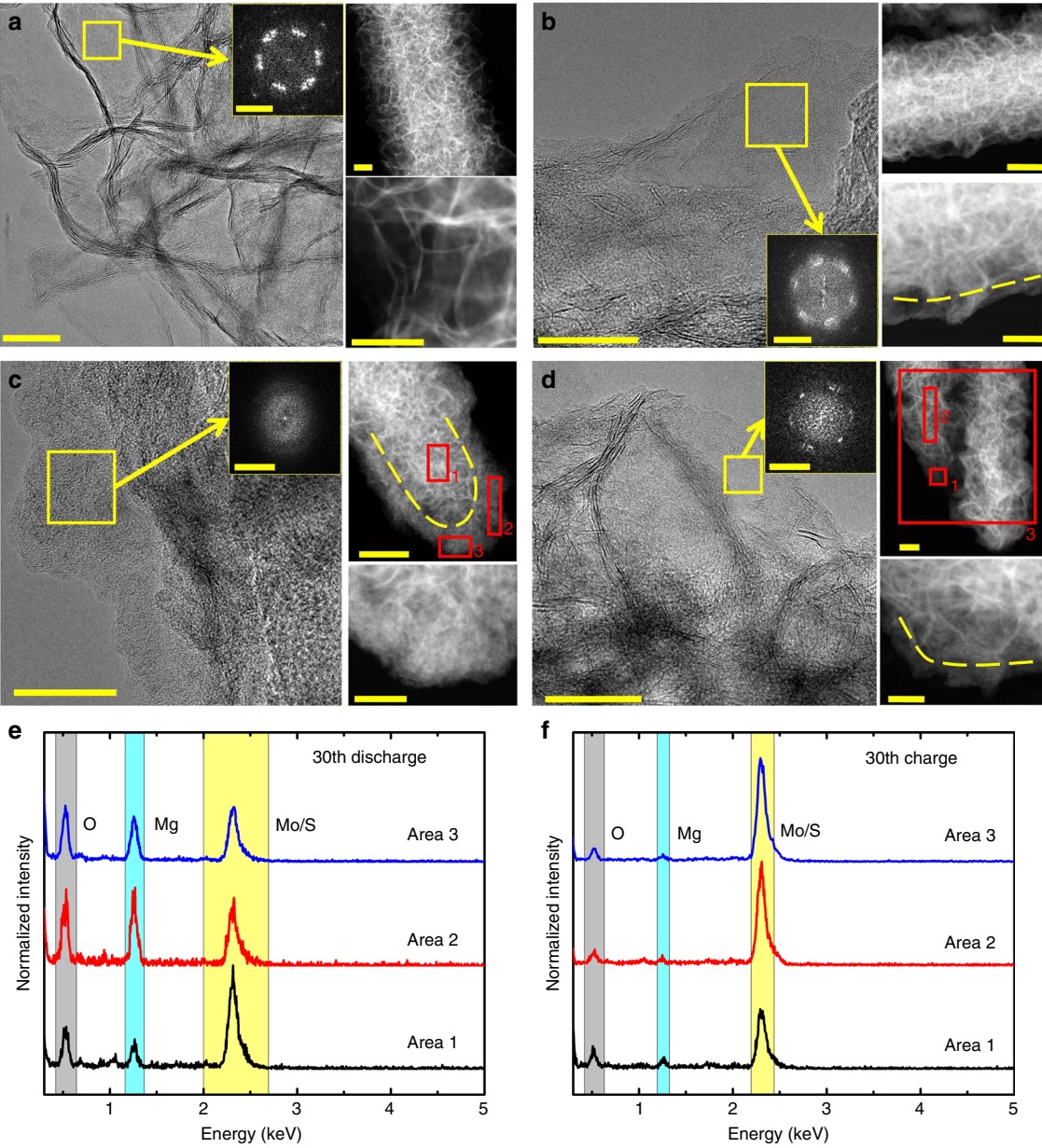

**Fig. 4** Ex situ HRTEM measurement of $MoS_2$@C-PNR electrodes. HRTEM image (left) with the FFT insert of the surface area and STEM-HAADF images (upper/lower) of $MoS_2$@C-PNR electrodes at **a** the pristine state, **b** 10th discharged state, **c** 30th discharged state, and **d** 30th charged state. The yellow dashed lines indicate the border between the activated and inactive area. The scale bar of HRTEM images corresponds to 20 nm, the FFT insets to 5 1/nm, upper STEM-HAADF images to 100 nm, and lower STEM-HAADF images to 50 nm. The corresponding EDX spectra of the indicated areas in the STEM-HAADF images of the 30th discharged and 30th charged state are shown in (**e**) and (**f**)

with the activation process. It should be mentioned that the capacity obtained is normalized based on the whole mass of the activated material, but in this case only the surface region is activated. Therefore, further downsizing the materials is a promising route for increasing the capacity. Interestingly, by comparing the O and Mg peaks in Fig. 4e, f, we notice the oxygen content varies simultaneously with the magnesium content during cycling. Considering the negligible O content in the pristine sample (Supplementary Fig. 2), the O signals cannot be attributed to contamination. Furthermore, the oxygen EDX maps indicate that O has the same distribution of Mg, which presents mainly in the area of the activated $MoS_2$ sheets close to the surface, but with less intensity in the unreacted core of the rod. To further confirm the spatial correlation between the O and Mg, principal component analysis (PCA) of the STEM-EDX maps of

both 10th and 30th discharge state was performed. The PCA results (Fig. 5b, d) indicate that two principal components can describe the whole EDX spectral images. One of the component spectra (red curve in Fig. 5b, d) presents only one pronounced peak at ~2.3 keV corresponding to the Mo-L/S-K peak, which indicates that S is still bonded to Mo rather than Mg. Another component spectrum (blue curve in Fig. 5b, d) exhibits two pronounced peaks for O-K and Mg-K signals, confirming a strong correlation of the spatial distribution of O and Mg. This would be expected for intercalation of $[Mg(DME)_x]^{2+}$. This is in agreement with our previous study[28], which showed that each $Mg^{2+}$ is coordinated by three DME molecules in the MgBOR/DME electrolyte. Considering the strong coordination of $Mg^{2+}$ by DME, it has been predicted that a high dissociation energy would be need to be overcome for pure $Mg^{2+}$ intercalation.

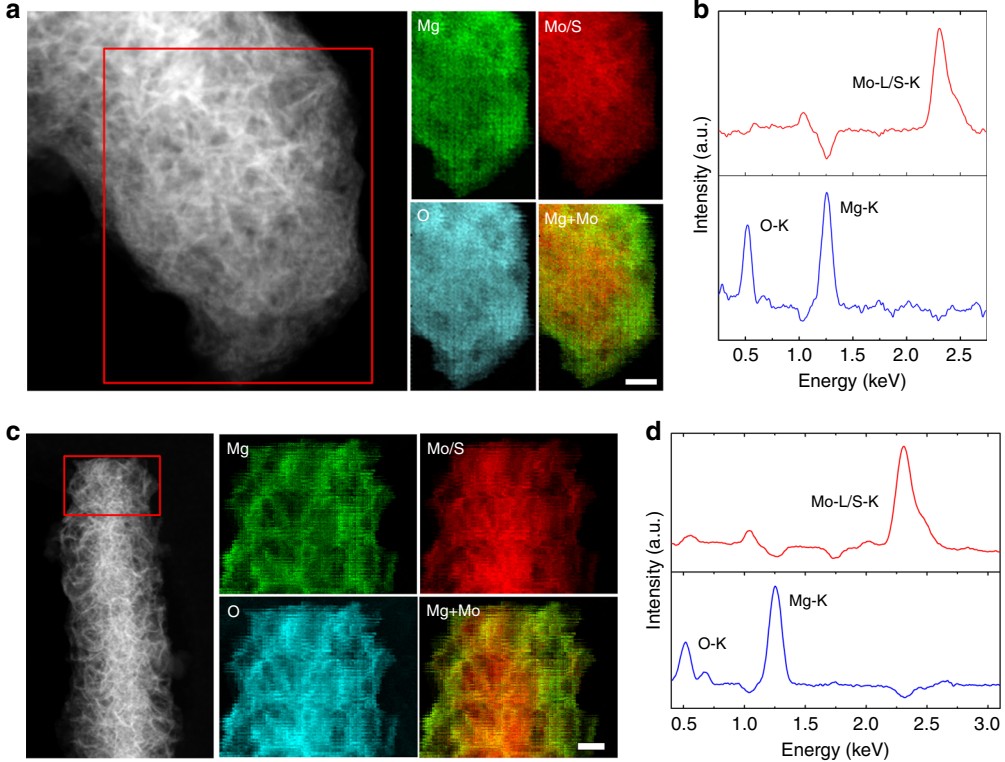

**Fig. 5** EDX mapping and PCA. STEM-HAADF image and the corresponding EDX maps of MoS₂@C-PNR electrodes at **a** 30th discharged state and **c** 10th discharged state taken from the area highlighted by red box in the corresponding STEM-HAADF image. The scale bars of element maps correspond to 50 nm. PCA components of the EDX spectral image of samples at **b** 30th discharge state and **d** 10th discharged state

The atomic structure of the dis-/charged samples was investigated by pair distribution function (PDF) analysis of the 4D-STEM diffraction data, which was taken from the activated shell, e.g., as shown in Supplementary Fig. 17. The PDF describes the probability of observing atomic pairs with a distance $r$ (with accuracy better than 0.02 Å) and hence reflects the short- and medium-range order. Figure 6a shows the PDFs of the dis-/charged samples (at 30th cycle) and the pristine sample. By comparing them with the simulated partial PDFs of MoS₂ with 2H- and 1T-structures[40] (Supplementary Fig. 18), we know that the peak at 2.4 Å, which does not change in the PDFs for all the three states, corresponds to the Mo–S bonding distance. The peak at 3.1 Å corresponds to distances of nearest (1st order) Mo–Mo and S–S. This peak significantly shifts lower in the discharge state comparing to the charged and pristine sample. Although that no detectable Mo–Mo and Mg–S direct bond can be found in the PDF of discharged state strengthens the intercalation mechanism, the shift of this 3.1 Å peak indicates a reduction of the Mo–Mo nearest neighbor distance, thus suggests Mo reduction in the discharged state. This change in structure is also responsible for the shift of the peak at 5.5 Å, which is attributed to the high order (medium-range) Mo–Mo pairs. For the discharged sample, it merges with the higher order Mo–S peak at ~5.2 Å, while it is restored after charging the sample. This atomistic interpretation fits to the HRTEM results, where the crystalline order of the MoS₂ layers is strongly reduced in the discharge state, while the ordering recovered after deintercalation of the Mg²⁺.

The PDF peaks at 4.5 and 6.3/6.7 Å (highlighted by red arrows in Fig. 6a), especially the ratio between the 6.3 and 6.7 Å peaks, are related to the S–S medium-range order; hence characterize the difference between H- or T-type structures as shown in Fig. 6a and Supplementary Fig. 18. These features in Fig. 6a indicate MoS₂ layers in the pristine and the activated MoS₂ sheets of the charged state are dominated by the H-type structure, while in the

discharged state, the short-/medium-range atomic structure of the nearly amorphous MoS₂ fragments possesses the T-type characteristics. The above results indicate a combined Mg storage mechanism, in which the intercalation is accompanied by the reversible 2H to 1T phase transition.

It is important to note that the hypothesis of solvated ions intercalation, based on the above EDX analysis, is corroborated by PDF analysis. In Fig. 6a, the peak at 1.4 Å corresponds to C–C and C–O bonds. The presence of C–C bonds is expected in the pristine sample due to the polydopamine derived carbon support of the MoS₂@C-PNR. This peak significantly increases in the discharged sample and reduces in the charged sample, suggesting carbonaceous groups related to Mg²⁺ ions to participate in the cycles. It agrees with the C map and corresponding PCA of the 30th discharged sample (Supplementary Fig. 19), which shows a strong correlation of C and Mg, O. The shielding effect of the solvent molecules is expected to be a reason why the Mg²⁺ mobility is enhanced, which could be another reason for the activation process. Furthermore, a noticeable peak at 2.0 Å appears at the left of the 1st Mo–S peak in the discharged PDF, but disappears in the charged one. To more clearly elucidate the short-range structural difference between the discharged and charged samples encoded in their PDFs, the dis-/charged PDFs are used as input for an independent component analysis (ICA)[41]. The ICA results in two solutions shown in Fig. 6b, denoted as IC1 and IC2. IC1 (blue solid) shows identical feature to the PDF of the charged state (blue dashed). IC2 (black solid) highlights the short-range order difference between the discharged and charged states. The simulated PDF of [Mg(DME)₃]² ⁺ clusters (Fig. 6c) is based on the structure reported in the literature[28]. For the simulation, we eliminated the atomic pairs with distances larger than 3.2 Å and O–C and Mg–C correlations, as the molecules are expected to be strongly distorted after the intercalation and as the high energy electron beam will partially

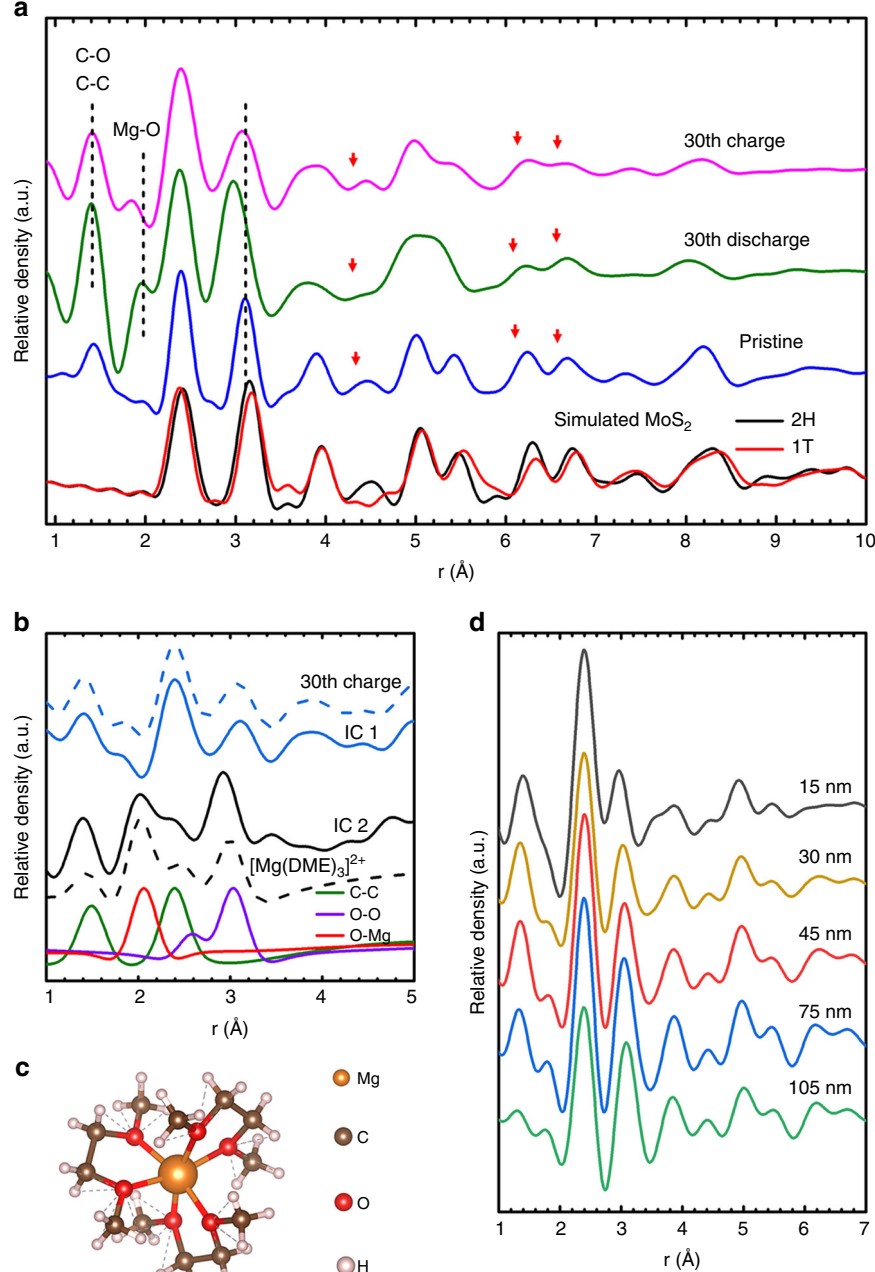

**Fig. 6** PDF analysis and depth profile. **a** PDF analysis of 4D-STEM diffraction patterns taken from the area highlighted in Supplementary Fig. 17. **b** ICA solution to the PDF curves and the simulated PDF of a $[Mg(DME)_3]^{2+}$ molecular cluster. **c** Molecular structure of a $[Mg(DME)_3]^{2+}$ cluster. **d** Local PDFs at a different depth from the surface of the 30th charged sample

break the O–C and C–H bonds. The IC2 fits well to the simulated PDF of the $[Mg(DME)_3]^{2+}$ cluster. Furthermore, comparison of the curves reveals that the shoulder at 2.0 Å in the discharged state in Fig. 6a corresponds to Mg–O bonds, which exist in the intercalated $[Mg(DME)_3]^{2+}$ cluster. In addition, the peak at 3.0 Å in IC2 fits to the length of the O–O pairs in $[Mg(DME)_3]^{2+}$. However, this distance is not unique and presumably mainly due to the reduced Mo–Mo distance in the discharged 1T-MoS₂. Nevertheless, overall these observations fit well to the hypothesis that the $[Mg(DME)_3]^{2+}$ cluster is intercalated into the MoS₂ from the electrolyte.

Figure 6d shows local PDFs in the 30th charged sample measured at a different depth from the surface. In the PDF measured at the top surface (<15 nm, black curve), the peak corresponding to the 1st order Mo–Mo distance is shifted from

the 3.1 Å in the 2H-type structure to shorter distance of 3.0 Å, and the peak height is reduced. As mentioned above, these changes indicate a Mo reduction. Furthermore, the peak at 4.5 Å is disappeared comparing that in the H-type structure. All these features fit well to the 1T-type structure and the PDF of the discharged state, whereas the PDFs measured for material located further than 30 nm from the surface still match the 2H-type structure. This finding indicates the top most surface layer (<30 nm) in the charged sample possesses a 1T-type structure with reduced Mo, although the majority of the activated material in the shell below the surface has been transferred back to the 2H-type structure after charging. The observation of this thin layer on the top surface fits to the STEM-HAADF image in Fig. 4d, where material less than 30 nm from surface in the charged sample still consists of amorphous fragments, indicated by the yellow dashed

line in Fig. 4d. This explains the XPS measurements (Fig. 3) of the charged state, which exhibits high 1T content with reduced Mo at a similar level as in the discharged state. At the surface, the 1T-type $MoS_2$ is stabilized by residual $Mg^{2+}$ even in the charged state. Some amount of the residual $Mg^{2+}$ was also confirmed by EDX (Supplementary Fig. 16b). The higher C–C peak in the average PDF (pink curve in Fig. 6a) and local PDFs of the activated material in the charged sample (black, yellow, red, and blue curves in Fig. 6d) compared to that of the pristine sample (blue curve in Fig. 6a) and the inactivated core of the charged sample measured at 105 nm from the surface (green curve in Fig. 6d) can be attributed to the presence of DME molecules which remain partially in-between the $MoS_2$ layers even at the fully charged state.

With the above results, we can explain the mechanism of the fast kinetics of the Mg intercalation into the $MoS_2$ with MgBOR/DME electrolyte. As shown in the schematic illustration in Fig. 7, in stage 1 to 2, the intercalation of $Mg^{2+}$ into the $MoS_2$ layers is limited by its low ion mobility as a result of the strong interaction between $Mg^{2+}$ and the host in the first several cycles. Instead, the solvated $[Mg(DME)_3]^{2+}$ ions, which are the stable form in the electrolyte, have a much larger volume per cation. In this case, $[Mg(DME)_3]^{2+}$ might have a lower intercalation energy barrier than $Mg^{2+}$, although the radius of the ion is larger. A recent report[42] showed that linear glymes (DEG/DME) are able to unlock the intercalation of $Mg^{2+}$ ions into graphite layers by cointercalation, which supports this point. However, with the large cation insertion, the interlayer will suffer from expansion and distortion, which creates lots of defects in each layer. This leads to a breakdown of the 2H-$MoS_2$ layer, resulting in fragmentation and structural distortion, which can further induce a phase transition from 2H to 1T phase[34]. As sketched in stage 3, the $[Mg(DME)_n]^{2+}$ ($n = 0$–3) cations are dispersed in the fragments of 1T-$MoS_2$. Since the ion mobility is limited, the phase transformation only takes place close to the surface of the rods. However, with the transition from semi-conductive 2H to

metallic 1T phase, the intrinsic conductivity is largely improved. With increasing fragmentation of the $MoS_2$ sheets, the electrochemical reactions speed up and the cation can insert deeper into the layered structures. While in the charge process shown in stage 4, the 1T-$MoS_2$ fragments prefer to rearrange and transform back to thermodynamically stable 2H phase when the $[Mg(DME)_n]^{2+}$ ions are extracted from the host. But the $MoS_2$ fragments formed in the discharge still keep their nanoscale, forming an active zone. After cation deintercalation, there are some DME molecules and a trace amount of $Mg^{2+}$ left in-between the 2H-$MoS_2$ layers. The remaining DME molecules also help to improve the kinetics because they act as the shielding layer to weaken the interactions between $Mg^{2+}$ and the host materials. Based on both XPS spectra and TEM images, the $Mg^{2+}$ residues are concentrated at the surface after charging. The remaining DME and $Mg^{2+}$ residues help to stabilize the 1T phase at the surface. With consecutive cycling, the active zone becomes larger and larger. Because of that the host material is capable of hosting more $Mg^{2+}$, resulting in an increasing capacity in the first few cycles. Based on the proposed mechanism, the electrochemical reactions could be summarized as follows:

$$MoS_2(2H) + x\left[Mg(DME)_n\right]^{2+} + 2x\,e^- \rightarrow \left[Mg(DME)_n\right]_x MoS_2(1T) \tag{1}$$

$$\left[Mg(DME)_n\right]_x MoS_2(1T) \leftrightarrow (DME)_y MoS_2(2H) + x\left[Mg(DME)_n\right]^{2+} + 2xe^- \tag{2}$$

where $[Mg(DME)_n]^{2+}$ refers to different solvated cations with $n = 0$–3, while $x$ stands for the amount of Mg stored per $MoS_2$ molecular unit.

The improvement of the conductivity and the $Mg^{2+}$ mobility are characterized by the potentiostatic electrochemical impedance spectroscopy (EIS) and the galvanostatic intermittent titration

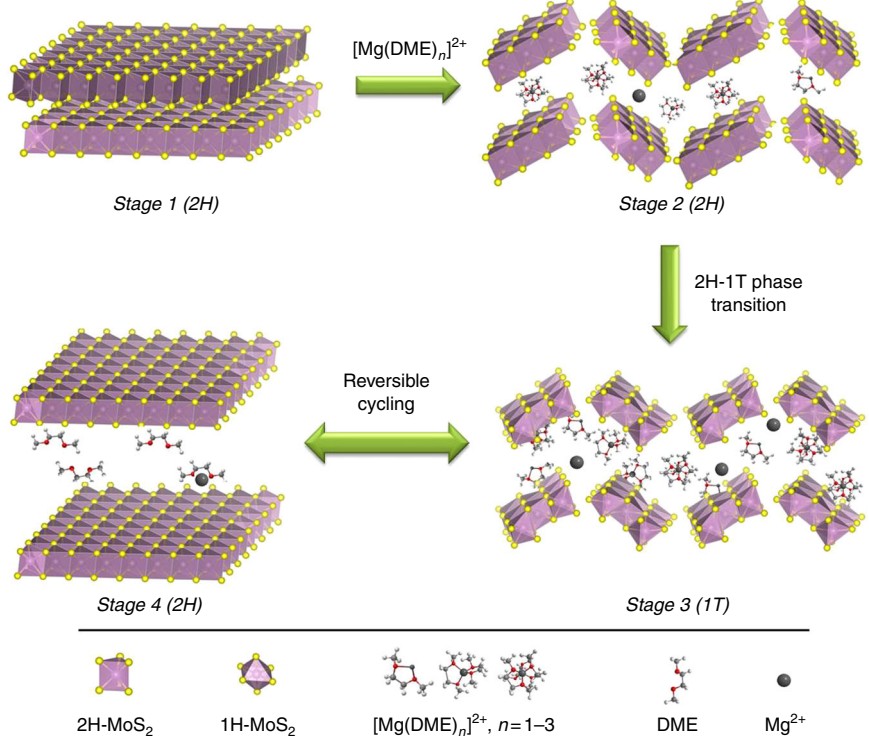

**Fig. 7** Schematic illustration of the Mg storage mechanism in $MoS_2$ structures with MgBOR/DME electrolyte

technique (GITT). As shown in Supplementary Fig. 20, the EIS of the pre-cycled battery shows a large semicircle in the whole frequency range, indicating a large resistance of the battery before cycling. But the semicircle becomes much smaller after the 1st cycle and reduces further in the following cycles as shown in Fig. 8a. The fitting of the EIS data based on the equivalent circuit in Supplementary Fig. 20c determines the resistance of anodes ($R_2$) and cathodes ($R_3$), both values decrease sharply from the 1st cycle ($R_2 = 873.6\ \Omega$, $R_3 = 866.3\ \Omega$) to the 30th cycle ($R_2 = 278.4\ \Omega$, $R_3 = 109.6\ \Omega$), indicating a largely enhanced ion and electron transfer. Moreover, the kinetics of the $Mg^{2+}$ diffusion in the MoS$_2$@C-PNR cathode was investigated by GITT measurement. As shown in Supplementary Fig. 21, the measurement is conducted at a constant current pulse of $20\ mA\ g^{-1}$ for 10 min followed by a relaxation period of 20 min. The $Mg^{2+}$ diffusivity was determined based on the equation in the Supporting Information. From Fig. 8b, the $Mg^{2+}$ storage is limited by its diffusivity in MoS$_2$. No more $Mg^{2+}$ could be stored when its diffusivity is below $\sim 10^{-12}\ cm^2\ s^{-1}$. In the 1st cycle, $Mg^{2+}$ diffusivity decreases quickly with the intercalation of the Mg-based cations resulting in only less than 0.2 $Mg^{2+}$ per MoS$_2$. Upon cycling, however, the value is largely improved due to the phase transition and the shielding effect of the remaining DME molecules in the MoS$_2$ layers. After 30 cycles, the cathode material is capable of storing more than 0.4 $Mg^{2+}$ which is more than double than that of the 1st cycle. For comparison, the $Mg^{2+}$ diffusivity in MoS$_2$-com cathode decreases sharply even at the 30th cycle (Fig. 8c), which might result from the much lower surface area of the bulk MoS$_2$ (MoS$_2$-com) compared to the nanosized MoS$_2$ (MoS$_2$@C-PNR) so that less MoS$_2$ is activated. Furthermore, the fast kinetics is also evident by a Randles–Sevcik analysis of the CVs with different scan rates (Supplementary Fig. 9). By fitting the current with a power law, the diffusion controlled and capacitive contribution of the capacity were determined (see Supplementary Fig. 22). The high capacitive contribution shown in Fig. 8d can be attributed to a prompt near-surface electrochemical reaction indicating a faster kinetics powered by the quick $Mg^{2+}$ mobility and fast electron transfer.

## Discussion

We present a divalent intercalation chemistry in RMBs with fast kinetics enabled by solvated $Mg^{2+}$ ions ([Mg(DME)$_3$]$^{2+}$) from a halogen-free non-corrosive electrolyte. With the solvated ions intercalation, the shielding effect of DME molecules in-between the $Mg^{2+}$ and the 2D host (MoS$_2$) layers weakens the interaction between the inserted ions and the lattice, resulting in a higher $Mg^{2+}$ mobility. The proposed mechanism of intercalating solvated $Mg^{2+}$ ions demonstrates a great potential of enhancing the multivalent ion battery performances by studying the compatibility between the cathode and the electrolyte. Moreover, the divalent ion intercalation induced a 2H to 1T phase transition in the MoS$_2$@C-PNR layered structure, resulting in the formation of a metallic 1T-MoS$_2$ structure in the discharged state, which is capable of fast electron transfer and helps to overcome the $Mg^{2+}$ diffusion energy barrier. The majority of the 1T phase MoS$_2$ reverses back in the charged state, only leaving a distorted 1T structure with fragmented morphology at the top surface of the electrode material. The reversible phase transition reduced the MoS$_2$ nanosheet size forming an activated zone, offering a further enhanced kinetics. As a result, the battery configuration exhibits a high capacity of $120\ mA\ h\ g^{-1}$ and a cycling stability for up to 200 cycles even at $0.5\ A\ g^{-1}$. The thorough mechanistic study reveals the reversible phase transition during $Mg^{2+}$ de-/intercalation and clarifies the Mg storage process in MoS$_2$ materials which is promising for many intercalation-based layered transition metal sulfides. The strategy of solvated $Mg^{2+}$ ion intercalation paves a new way to overcome the sluggish intercalation kinetics in multivalent ion-based battery systems.

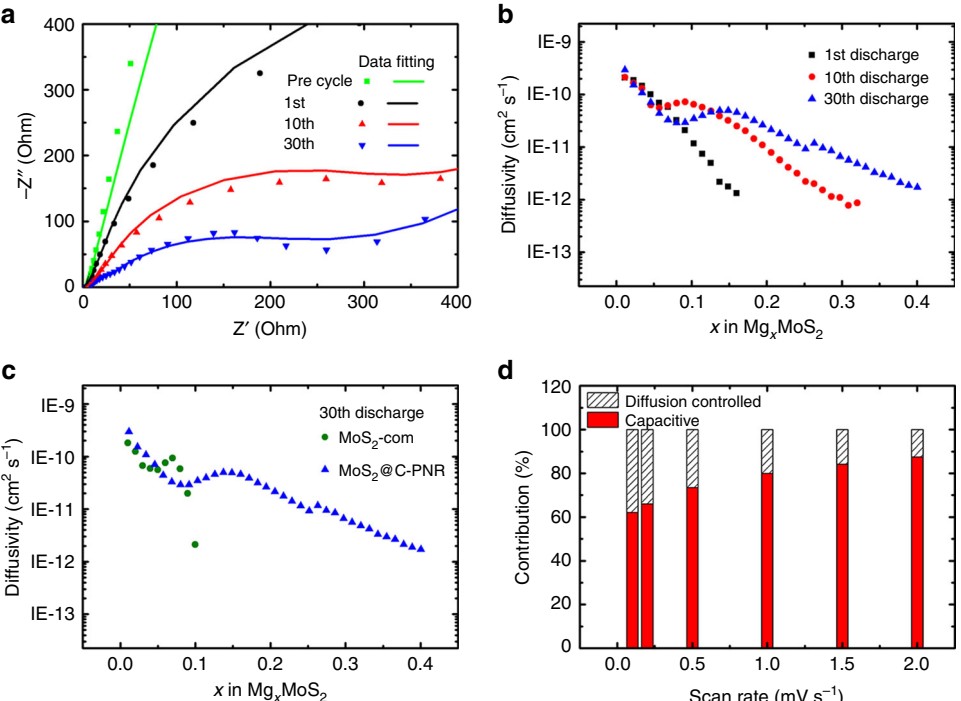

**Fig. 8** Kinetic study based on electrochemical measurements. **a** EIS of MoS$_2$@C-PNR at electrodes after specific cycles in the high to medium frequency range. **b** $Mg^{2+}$ diffusivity of MoS$_2$@C-PNR during magnesiation process based on GITT. **c** Comparison of the $Mg^{2+}$ diffusivity in MoS$_2$@C-PNR and MoS$_2$-com at the 30th discharge. **d** Capacitive and diffusion controlled contributions of MoS$_2$@C-PNR in the electrochemical processes at different scan rates

## Methods

**Materials synthesis**. MoS$_2$@C porous nanorods (MoS$_2$@C-PNR) were prepared by a simple self-sacrificial template method in which α-MoO$_3$ nanorods were used as a template. Typically, the α-MoO$_3$ nanorods were synthesized by treating ammonium molybdate tetrahydrate ((NH$_4$)$_6$Mo$_7$O$_{24}$·4H$_2$O, Sigma-Aldrich, 81.0–83% MoO$_3$ basis) hydrothermally in a 3 M nitric acid solution at 180 °C for 30 h[43]. The resulting white powder was mixed with dopamine hydrochloride in a tris(hydroxymethyl)aminomethane solution under vigorous stirring overnight. Then thiourea was added to the above solution and stirred for another 1 h. The molar ratio of Mo and S from the precursors was 1:5. After complete dissolution of thiourea, the solution was transferred into a 125 mL Teflon autoclave and reacted at 200 °C for 24 h. The final product was obtained by pyrolysis of the black powder from the hydrothermal reaction in a SS reactor under Ar at 600 °C for 3 h.

**Characterization**. XRD was performed using a Bruker-AXS D8 diffractometer in Bragg–Brentano geometry by applying Cu-K$_{\alpha 1}$ radiation (λ = 1.541 Å) with a step size of 0.02°. Thermal analysis of the samples was carried out with TGA coupled with differential scanning calorimetry in a Setaram thermal analyser of a SENSYS evo instrument. The measurement was conducted from room temperature to 700 °C under synthetic air flow with a heating rate of 5 °C min$^{-1}$. SEM images were obtained using a ZEISS LEO 1530 at 15 kV. The SEM samples were prepared on carbon tape followed by gold sputtering. XPS measurements were carried out in a PHI 5800 MultiTechnique ESCA system (Physical Electronic) using monochromatic Al K$_\alpha$ (hv = 1486.6 eV) radiation (250 W, 13 kV), at a detection angle of 45°, and with pass energies at the analyser of 93.9 and 29.35 eV for survey and detail scans, respectively. The C 1s line of the conductive C additive in the electrodes (see below) was used as binding energy (BE) reference and set to 284.6 eV. The spectra in the Mo 3d and S 2p range were analyzed by peak fitting using CasaXPS (Shirley background, Gaussian/Lorentzian peak shape). For peak fitting, two constraints were kept in the deconvolution: (1) the intensity ratio of the two species was the same in the S 2p and Mo 3d spectra; (2) the BE distance for the peaks of the two species was fixed for all samples. It may be noted that the S 2s peak also appears in the Mo 3d range; the S 2s peak of MoS$_2$ was fixed at 226.8 eV, its intensity was determined by comparison to the intensity of the S 2p peak (taking into account the relative sensitivity factors of the two sulfur peaks). The S/TEM studies were performed using an aberration (image) corrected FEI Titan 80–300 microscope operated at 300 kV, equipped with a Gatan UltraScan CCD camera and an EDAX S-UTW EDX detector. For all ex situ measurements including XRD, XPS, and TEM, the samples were collected from the cathode of the cells at given dis-/charge states. They were thoroughly washed with anhydrous DME in an Ar-filled glovebox and dried in vacuum at 80 °C overnight. The ex situ XRD measurements were performed under protective Ar atmosphere. The TEM samples were transferred under Argon from the glovebox to the microscope using a Gatan 648 vacuum transfer holder minimizing air contamination. Similarly, an inert gas transfer system (PHI model 04-110) was used for transport to the XPS spectrometer.

The STEM-EDX spectral images were analyzed by PCA[44]. It examines the covariance matrix of the spectra of the STEM-EDX map and reveals the spatial correlation of elements in the material. Diffraction data was acquired in microprobe STEM diffraction (4D-STEM) mode using almost parallel nanobeam conditions. The convergent semi-angle was 0.6 mrad and the corresponding size of the electron probe was 1.7 nm. Local atomic PDFs were calculated from the 4D-STEM diffraction patterns according to the method described in refs. [45,46].

**Electrochemical measurement**. Electrochemical measurements were carried out using a Swagelok cell configuration except the CV, which was conducted using a PAT-Cell configuration from EL-Cell. The cathode was made by coating SS foil with a slurry of 70% active material (MoS$_2$@C-PNR or commercial MoS$_2$ powder), 20% Super P (Timcal) and 10% Solef polyvinylidene difluoride (PVDF) in N-methyl-2-pyrrolidone (NMP) followed by vacuum drying at 80 °C for 15 h. A typical mass loading of the active material was 1.5 mg cm$^{-2}$. The magnesium tetrakis(hexafluoroisopropyloxy)borate (MgBOR) electrolyte was synthesized in a one-pot reaction between Mg(BH$_4$)$_2$ and hexafluoroisopropanol in DME following our previous work[28]. The cells were assembled in an Ar-filled glovebox (H$_2$O and O$_2$ < 0.5 ppm) using 0.4 M MgBOR in DME as electrolyte, Mg foil as counter electrode, and glass fibre as separator. For a three-electrode cell, Mg foil was also used as reference electrode. CV and EIS measurements were performed using a VMP3 multichannel potentiostat (Bio-Logic SAS). The EIS data were analyzed with the help of the Z Fit function of the EC-Lab software. Other measurements including galvanostatic cycling and GITT were carried out using an Arbin battery cycling unit. All the electrochemical measurements were conducted at 25 °C except particularly mentioned.

The other electrolytes used in this work were prepared as follows: The APC electrolyte was prepared by diluting a 2.0 M phenylmagnesium chloride/tetrahydrofuran solution with tetrahydrofuran (THF) followed by dissolving one equivalent of AlCl$_3$ in it. The Mg(TFSI)$_2$ electrolyte was prepared by dissolving magnesium bis(trifluoromethanesulfonimide) in acetonitrile and stirring overnight before use. The HMDS electrolyte was prepared by first dissolving magnesium bis (hexamethyldisilazide) and two equivalents of AlCl$_3$ in a diglyme/tetraglyme (1:1 v/v) mixture and stirring overnight. Then 0.5 equivalents of MgCl$_2$ were added

followed by stirring for another 48 h. The Mg concentration in all the electrolytes was kept at 0.4 M.

## Data availability

All experimental data that support the findings of this study are readily available upon request of the corresponding authors.

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

## Acknowledgements

This study is supported by the "MagS" project (03XP0032A) from Bundesministerium für Bildung und Forschung (BMBF) of Germany. X.M. would like to acknowledge the funding by Deutsche Forschungsgemeinschaft's grant MU 4276/1-1 for supporting the development of the 4D-STEM PDF. This work contributes to the research performed at CELEST (Center for Electrochemical Energy Storage Ulm-Karlsruhe). We acknowledge support by Deutsche Forschungsgemeinschaft and Open Access Publishing Fund of Karlsruhe Institute of Technology.

## Author contributions

Z.L. synthesized the materials, fabricated the electrodes, and performed all the electrochemical measurement. X.M. and C.K. conducted the STEM measurement and did the related analysis. Z.Z.-K. synthesized the electrolyte. T.D. and R.J.B. conducted the XPS measurement and analyzed the data. M.F. directed the project. The manuscript was written through contributions of all authors. All authors have given approval to the final version of the manuscript.

## Additional information

**Competing interests:** The authors declare no competing interests.

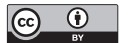

