## [Peer Review File · Nature Communications]

Reviewers' comments:

Reviewer #1 (Remarks to the Author):

Comments to the Authors of NCOMMS-18-22910

The manuscript reports a detailed analysis of magnesium intercalation into a molybdenum sulfide cathode for magnesium secondary batteries. Even if the battery performances are not outstanding, the authors did a great job in analyzing and revealing how the Mg^{2+} intercalation process occurs. This is a very good paper, and it will be very helpful for other scientists working in this field. Indeed, the second part of the manuscript is very detailed, and results are really interesting. I have a very positive opinion about the different approaches used in this work, and the conclusions derived therefrom. This manuscript should be reconsidered for publication in Nature Communications only after a minor revision.

Here below are listed all the points that should be reconsider before the acceptance of the manuscript:

1. Page 7 lines 1-7: the XRD paragraph is poorly described. However, with this technique, useful information can be obtained. Did the authors tried to perform Rietveld analysis? Are the crystal sizes along the different directions coherent with SEM and TEM results?
2. Page 9 Electrochemical Performances paragraph: the authors used a 0.4 M MgBOR in DME electrolyte. Even if this electrolyte shows very interesting features, especially in terms of reversibility and ESW, it reveals a quite large overpotential between Mg deposition and dissolution (i.e. ca. 0.7 V). This results in a low energy efficiency in the battery tests. Is this the best electrolyte for magnesium deposition and dissolution in terms of overpotential? Would it be interesting to check if better battery test results can be obtained using other electrolytes?
3. Page 10 Cyclic Voltammetry experiments: are the tests conducted using a three- or two-electrodes measuring cell? CV tests should be carried out using a three-electrodes cell in order to obtain more precise results.
4. Page 10 Galvanostatic Cycling results: despite the very high capacity values reported in this manuscript, one has to keep in mind that the devices are working at a very low potential. Only the 25 % of the capacity is above 1.0 V (see Fig. 2b), that corresponds to ca. 30 mAh·g⁻¹. These results are far from be used in a real application.
5. Page 10 Galvanostatic Cycling results: what are the specific energy values obtained by this material? And furthermore, what are the energy efficiency values reached by these devices? Coulombic efficiency is very important, but in real applications also the energy efficiency values have to be considered.
6. Page 12 Fig. 3e: are these structural fractions calculated on Mo 3d or S 2p fitting results? Or is it an average of the two? Do they give the same result?
7. Supporting Information page 8 Fig. S12c: is this equivalent circuit in agreement with others reported in literature? Are the attributions of the single components of the equivalent circuit to the physical events occurring in the testing device correct?

MINOR POINTS:

1. Page 4 line 31: I guess typo, $MgCl^+$, not $MgCl^-$.
2. Page 9 Fig 2a: if "Fast Kinetics" are claimed, why CV tests are carried out at 0.1 mV·s⁻¹? Should be results at faster rates (Fig. S14a) discussed in this paragraph, instead of at the end of the manuscript?
3. Page 12 Fig. 3: the legend is incorrect. (a) is the dis-/charge profile; (b) ex situ XPS spectra; (c) and (d) MoS₂ structures; (e) fractions of MoS₂ components. Fig. 3b, S 2p spectra: from charge state

D to G, the fitting is not so good respect to the experimental results. Can it be improved? Fig. 3b, charge state A, Mo 3d spectra: is the green color of the low intensity curve correct? Or should it be red, corresponding to the 1T- MoS₂? Fig. 3b, charge state G, Mo 3d spectra: why the fitting curve in this case is colored? It should be black.

4. Pages 12 and 13: all the numbers of Fig. 3 reported in the text should be checked and updated with the new corrected legend of Fig. 3.

5. Page 22 Fig. 8b and 8c: what are the differences between these two figures and Fig. S13c and S13d?

6. Page 22 Fig. 8d: what is the difference between this figure and Fig. S14d?

7. Supporting Information: if this is the final version of the Supporting Information, figures and corresponding figure captions should stay in the same page.

8. Supporting Information page 2 Fig. S1: units should be reported into brackets to be coherent with all the other graphs. Furthermore, what is ZL043 reported in the figure caption? I guess it is MoS₂@CPNR.

Reviewer #2 (Remarks to the Author):

Title: Fast kinetics of multivalent intercalation chemistry enabled by solvated Mg²⁺ ions into self-established metallic layered materials

Authors: Zhenyou Li et al.

This paper claims facile intercalation of solvated Mg²⁺ ions in metallic 1T-phase MoS₂. The claimed novelty includes (1) intercalation of solvated Mg²⁺ to reduce electrostatic interaction between Mg²⁺ and MoS₂ slabs and (2) self-establishment of metallic 1T-phase by multivalent ions for the first time. However, substantial level of intercalation of solvated Mg²⁺ has been reported earlier with different intercalation host, namely graphite (Chem. Mater. 2018, 30, 3199-3203). Therefore, the novelty of this paper can be found in the electrochemical in-situ phase transition of MoS₂ from 1H to 1T phase. The reviewer cannot but question if this can be of interest to others in the community in the wider context. Perhaps, the novelty may be found by enabling those two claims in an electrolyte compatible to Mg metal anode. However, the first claim of "intercalation" seems not completely convincing, and following further evidences are required to strengthen the conclusion.

(1) The elemental ratio of Mg:Mo:S:O should be clarified based on EDX and XPS, respectively. Ideally, ICP can be a quantitative elemental analysis of choice.

(2) In the EDX mapping, can the authors clarify the C elements also?

(3) What is the role of C in the MoS₂/C composite? Please provide a data that rationalize the exclusion of C as a host for charge storage. For example, what happens for MoS₂/C sample with increased composition of C? Or C itself as an electrode?

(4) What is the redox center that counter-balance the storage of solvated Mg²⁺, especially if the oxidation number of Mo remains the same during charge-discharge?

(5) Any evidence from XANES, apart from XPS, which is too much surface-sensitive?

(6) The schematic figure 7 is problematic, because 1T phase persists according to XPS.

(7) Equation 2 is also problematic. Is there any evidence for the removal of Mg²⁺ leaving DME solvents in the structure? For example, EDX of charged status shows simultaneous removal of O and Mg from the structure.

(8) On the term "intercalation." Is it really intercalation, given that it is surface reaction according to CV, without any reversible changes in the structure and oxidation number?

(9) Can the authors provide similar experimental evidences on MoS₂-com, which can lead to clearer experimental observations due to the simplicity of the system, though the capacity is smaller?

(10) On the claim of highly reversible deposition and stripping of Mg in Fig. S3. What is the coulombic efficiency from the CV? Is it close to 100%?

Overall, I do not feel that the paper will influence or facilitate innovative thinking in the field. Also, I feel that their claims of solvated intercalation require more strengthened experimental evidences. Some part of the manuscript may need proof-reading for clarity in terms of English grammar. On the other hand, I find positive sides of this work that suggests a general concept of utilizing a host material via co-intercalation of solvated Mg^{2+} in an Mg metal-compatible electrolyte. However, that still requires more convincing study on the charge storage mechanism. Because of these reasons, I recommend to postpone the publication of this manuscript until the authors provide more detailed examinations on the interaction between of solvated Mg^{2+} and MoS_2/C composite and more deeper understanding of the charge storage mechanisms via experimental or theoretical approaches.

Reviewer #3 (Remarks to the Author):

In the present manuscript, the authors successfully demonstrated reversible co-insertion / co-desertion of solvated Mg^{2+} ion into $MoS_2@C$ -PNR. The authors conducted series of analytical study to understand the charge/discharge process of the $MoS_2@C$ -PNR. Even though the concept of co-intercalation itself is not very new, the co-intercalation of organic solvent for Mg battery is very unique. Also the manuscript is well written. Therefore I think overall the present manuscript is deserved for publication in Nature Communications. However couple of the points below must be revised before publication.

1. Even with the thorough discussions concerning the charge/discharge mechanism, the proposed mechanism in Fig. 7 is still doubtful, because all the XRD patterns shown in Fig. S6 have peaks at 33° , 58° . If the MoS_2 layer is broken down into the small pieces during the discharging process: 2H-1T phase transition shown in Fig. 7, these peaks are supposed to disappear at the discharged state. At first the authors should reconfirm the peak assignment of these XRD peaks. Then, an alternate charge/discharge mechanism should be proposed. I think this point is very critical for this paper.

2. The coulombic efficiency of the cell operated at 500 mA g^{-1} is highest among all the cells operated at different current density. What is the potential side reaction during the slow charge/discharge process?

3. Discussions concerning the simulated PDF in Fig. S11 sounds strange to me. The author assigned the peak corresponding to (1st order) Mo-Mo and S-S at 3.2 \AA shifts lower in the discharged state. However the Mo-S bond at 2.4 \AA does not change. It shows shrinkage of the lattice constant along the a, b axis and expansion of the c-axis. It does not match with the XRD patterns in Fig. S6, which does not show any peak shift.

4. The charge-discharge performance of the $MoS_2@C$ -PNR at elevated temperature should be presented in the manuscript, because the authors concluded that the diffusivity of Mg^{2+} limits the capacity. Of course I understand that the concept of the present work is the novel cathode at ambient temperature, but the capacity at the elevated temperature provides good target for the future works.

Response to the reviewers' comments

We greatly appreciate the reviewers' constructive comments. They offer us a good opportunity to carefully evaluate our data and review the interpretation again, which really helped us to gain a better overview and better understanding of the mechanism. Below are our answers and corresponding revisions to the reviewers' comments.

Reviewer #1:

The manuscript reports a detailed analysis of magnesium intercalation into a molybdenum sulfide cathode for magnesium secondary batteries. Even if the battery performances are not outstanding, the authors did a great job in analyzing and revealing how the Mg²⁺ intercalation process occurs. This is a very good paper, and it will be very helpful for other scientists working in this field. Indeed, the second part of the manuscript is very detailed, and results are really interesting. I have a very positive opinion about the different approaches used in this work, and the conclusions derived therefrom. This manuscript should be reconsidered for publication in Nature Communications only after a minor revision.

Here below are listed all the points that should be reconsider before the acceptance of the manuscript:

1. Page 7 lines 1-7: the XRD paragraph is poorly described. However, with this technique, useful information can be obtained. Did the authors tried to perform Rietveld analysis? Are the crystal sizes along the different directions coherent with SEM and TEM results?

According to this comment, we tried our best to conduct the Rietveld refinement and included it into the revised manuscript (Fig. 1a), even though it is quite challenging as the diffraction peaks are broadened and overlapped due to the distorted crystallinity of the rod-like structure which is an assembly of the MoS₂ nanosheets. The average crystal size along (100) and (110) directions were ~5 nm obtained by using Scherrer equation. The values match well with the electron microscopy results, where the (002) lattices of the MoS₂ nanosheets in the HRTEM image (Fig. 1f) can be seen to be strongly distorted. The ordered domains are much less than ~10 nm. The discussion of XRD data in the main text has been revised. A description of the refinement and the crystal size comparison between XRD and electron microscopy was also added.

2. Page 9 Electrochemical Performances paragraph: the authors used a 0.4 M MgBOR in DME electrolyte. Even if this electrolyte shows very interesting features, especially in terms of reversibility and ESW, it reveals a quite large overpotential between Mg deposition and dissolution (i.e. ca. 0.7 V). This results in a low energy efficiency in the battery tests. Is this the best electrolyte for magnesium deposition and dissolution in terms of overpotential? Would it be interesting to check if better battery test results can be obtained using other electrolytes?

We fully agree with the reviewer's opinion. We also noticed the relatively large overpotential in our cells. In fact, the 0.7 V overpotential observed in MoS₂@C-PNR|MgBOR|Mg cell (Fig.

2a of the previous version) are contributed by both the cathode and the anode. The overpotential of the cathode was determined to be 0.54 V from the three-electrode CV measurement (Fig. 2a) while the rest is from the anode as shown in Fig. S5. The low polarization of the electrolyte has been reported in our previous work (*ACS Energy Lett.* 3, 8, 2005-2013), in which a Mg-Mg symmetric cell shows only ~0.1 V overpotential. In order to highlight the high performance of our electrolyte, we made a comparison of the most commonly used electrolytes in MBs. From Tab. S1, it becomes obvious that our electrolyte is one of the best regarding overpotential.

Actually, the goal of this work is to improve the kinetics of de-/intercalation in MBs, aiming at reducing the overpotential as well. We were also trying to compare our full cell overpotential with other MoS₂-based cathode in MBs. However, most of the reported MoS₂-based cathodes (*Journal of Power Sources* 2017, 340, 104-110; *Nano Letters* 2015, 15(3), 2194-2202; *J. Mater. Chem. A* 2013, 1 (19), 5822-5826) exhibited only slope behavior in their charge-discharge profiles, which makes it difficult to compare.

As the reviewer suggested, we checked the battery performance of our cathode using the commonly used electrolytes, namely APC, HMDS, Mg(TFSI)₂. The results indicate that the MoS₂@C-PNR electrode in MgBOR electrolyte exhibits much better performance than in other tested electrolytes in terms of both capacity and energy density.

3. Page 10 Cyclic Voltammetry experiments: are the tests conducted using a three- or two-electrodes measuring cell? CV tests should be carried out using a three-electrodes cell in order to obtain more precise results.

The CV presented in the manuscript was conducted using a two-electrode Swagelok cell setup. Following the reviewer's suggestion, we conducted the three-electrode CV measurement, and replaced the two-electrode CV with the new data in the revised manuscript (Fig. 2a and Fig. S5). We plot the potential vs. Mg reference and Mg counter electrode separately. Both curves exhibit similar profile with two pairs of redox peaks. The main difference is the shift of the oxidation peak which is caused by the overpotential of the Mg anode. This overpotential also causes the ~0.2 V difference at both sides of the cut-off potential.

4. Page 10 Galvanostatic Cycling results: despite the very high capacity values reported in this manuscript, one has to keep in mind that the devices are working at a very low potential. Only the 25 % of the capacity is above 1.0 V (see Fig. 2b), that corresponds to ca. 30 mAh·g⁻¹. These results are far from be used in a real application.

We appreciate the reviewer's kind reminder. Accordingly we carefully rephrased our statement in the manuscript by highlighting the proposal and proof of the new intercalation concept rather than practical applicability of the materials. As it is known that the Mg-Ion intercalation into oxides suffers from extremely slow kinetics and sometimes undergoes irreversible conversion reactions. Use of "soft" chalcogenides would mitigate the electrostatic forces within the host matrix facilitating the intercalation. In this work, we chose MoS₂ as the model cathode to study its intercalation chemistry with the new Mg electrolyte. Nevertheless, the capacity above 1.0 V (about 30 mA h g⁻¹) in this work is higher than the state-of-the-art (< 20 mA h g⁻¹ in both *Journal*

of Power Sources 2017, 340, 104-110 and Nano Letters 2015, 15(3), 2194-2202). Encouraged by the discovery of the solvated-ion intercalation and the improved performance, our next step is to explore the cathode materials with potentially high capacity as well as high voltage.

5. Page 10 Galvanostatic Cycling results: what are the specific energy values obtained by this material? And furthermore, what are the energy efficiency values reached by these devices? Coulombic efficiency is very important, but in real applications also the energy efficiency values have to be considered.

According to this question and comment, we added the specific energy value and the energy efficiency into the revised version. Furthermore, we rephrased the statement to focus on proving the new intercalation chemistry for MBs. The concept could be applied to a broad range of layered materials towards a real application in the future. The low energy density and efficiency are mainly caused by the large overpotential and low potential plateaus as the reviewer pointed out before. For better answering reviewer's question, we probed our MoS₂ material with different electrolytes (Fig. S12). Thanks to the solvated ion intercalation, the cell with MgBOR electrolyte showed improved energy density.

6. Page 12 Fig. 3e: are these structural fractions calculated on Mo 3d or S 2p fitting results? Or is it an average of the two? Do they give the same result?

For XPS fitting, we keep two constraints of the deconvolution: (1) the intensity ratio of the two species is the same in the S2p and Mo3d spectra; (2) the binding energy distance for the peaks of the two species is kept constant for all samples. In this condition, the fractions calculated on Mo3d and S2p give the same result. This is why we did not label the fractions in Fig. 3e. We have modified the description of these constraints in the experiment part.

7. Supporting Information page 8 Fig. S12c: is this equivalent circuit in agreement with others reported in literature? Are the attributions of the single components of the equivalent circuit to the physical events occurring in the testing device correct?

The simulation of the EIS measurement referred to our previous studies of the electrolyte and other MBs systems using different electrolyte. We are sorry for the lack of the description of. The description of the equivalent circuit and the attributions of each element have been given in the revised supporting information. The equivalent circuit agrees with the previous reported system and could well explain the electrochemical process in our setup.

MINOR POINTS:

1. Page 4 line 31: I guess typo, MgCl⁺, not MgCl⁻.

We appreciate reviewers' careful reading. The typo has been corrected in the revised version.

2. Page 9 Fig 2a: if “Fast Kinetics” are claimed, why CV tests are carried out at 0.1 mV·s⁻¹? Should be results at faster rates (Fig. S14a) discussed in this paragraph, instead of at the end of the manuscript?

We would like to thank the reviewer for the precious suggestion. We have put the CVs with different scan rates in the electrochemical performance part and supplement the discussion there.

3. Page 12 Fig. 3: the legend is incorrect. (a) is the dis-/charge profile; (b) ex situ XPS spectra; (c) and (d) MoS₂ structures; (e) fractions of MoS₂ components. Fig. 3b, S 2p spectra: from charge state D to G, the fitting is not so good respect to the experimental results. Can it be improved? Fig. 3b, charge state A, Mo 3d spectra: is the green color of the low intensity curve correct? Or should it be red, corresponding to the 1T- MoS₂? Fig. 3b, charge state G, Mo 3d spectra: why the fitting curve in this case is colored? It should be black.

We appreciate the reviewer’s comments. The legend in Fig. 3 has been updated.

Regarding the fitting of XPS peaks, since we have to keep the intensity ratio of the two species the same in the S2p and Mo3d spectra, it is challenging to get much improvement. However, we still tried to improve the fitting by changing the spread in the S2p region from 0.4 to 0.5 eV. The updated fitting curves match to some extent better with the experimental data than the previous one.

The fitting of each peak was conducted in the whole range presented. Therefore, the green color of the low intensity in Fig. 3b charge state A is actually the extended fitting line of the neighbor peak. For other charge states, the green curves at lower intensities were also presented but covered by the 1T phase data (red curves). For the sample at OCV (charge state A), the 1T phase was not set in the fitting.

In the updated figure, the mistakenly colored curve is corrected.

4. Pages 12 and 13: all the numbers of Fig. 3 reported in the text should be checked and updated with the new corrected legend of Fig. 3.

The numbers of Fig. 3 reported in the text has been updated with the revised figure.

5. Page 22 Fig. 8b and 8c: what are the differences between these two figures and Fig. S13c and S13d?

They are the same figures. In the revised manuscript, the repeated figures have been deleted.

6. Page 22 Fig. 8d: what is the difference between this figure and Fig. S14d?

They are the same figures. The repeated figure was removed in the revised version.

7. *Supporting Information: if this is the final version of the Supporting Information, figures and corresponding figure captions should stay in the same page.*

This issue has been carefully addressed in the revised version.

8. *Supporting Information page 2 Fig. S1: units should be reported into brackets to be coherent with all the other graphs. Furthermore, what is ZL043 reported in the figure caption? I guess it is MoS₂@CPNR.*

Thank you for careful reading. The mistakes in Fig 1 (Fig. S2 in the revised version) have been corrected.

Reviewer #2:

This paper claims facile intercalation of solvated Mg²⁺ ions in metallic 1T-phase MoS₂. The claimed novelty includes (1) intercalation of solvated Mg²⁺ to reduce electrostatic interaction between Mg²⁺ and MoS₂ slabs and (2) self-establishment of metallic 1T-phase by multivalent ions for the first time. However, substantial level of intercalation of solvated Mg²⁺ has been reported earlier with different intercalation host, namely graphite (Chem. Mater. 2018, 30, 3199-3203). Therefore, the novelty of this paper can be found in the electrochemical in-situ phase transition of MoS₂ from 1H to 1T phase. The reviewer cannot but question if this can be of interest to others in the community in the wider context. Perhaps, the novelty may be found by enabling those two claims in an electrolyte compatible to Mg metal anode. However, the first claim of “intercalation” seems not completely convincing, and following further evidences are required to strengthen the conclusion.

We are grateful for referring the very recently published work on the incorporation of Mg²⁺ and linear ether into graphite (Chem. Mater. 2018, 30, 3199-3203), which we overlooked. This is a nice paper to clarify the cointercalation from both theoretical and experimental point of view. We have cited this paper in the revised version and rephrased some of our claims.

Here we would like emphasize that both works have different concepts and are targeting at different problems although share the same approach. The paper mentioned by the reviewer focused on solving the thermodynamic problem of intercalating Mg²⁺ ions to graphite anode. With the solvated ion intercalation, a ternary graphite interlayer compound was formed. But our work is aiming at building a general approach to improve the kinetics of the Mg²⁺ ion intercalation into layered cathodes. Furthermore, due to the cations (Mo⁴⁺) and anions (S²⁻) in the MoS₂ host, the electrostatic interaction between the host and the Mg²⁺ would be significantly larger than that between the monatomic graphite and Mg²⁺. This strong interaction is the main reason for the sluggish kinetics for intercalation type MB cathodes. Taking into account that most of electrode materials are chemical compounds rather than simple substances, the

mechanism reported in our work would be very interesting and can be potentially extended to other systems. In this light, we expect a remarkably broad readership of the current work.

1. The elemental ratio of Mg:Mo:S:O should be clarified based on EDX and XPS, respectively. Ideally, ICP can be a quantitative elemental analysis of choice.

According to this comment, the Mo:S:Mg ratio from XPS at each charge states has been added to the supporting information (Fig. S15). The stable Mo/S ratio indicates the unchanged chemical composition of the electrode material. The big difference of the Mg/Mo ratio at dis-/charge states and gradual increase with cycling suggest the Mg ions de-/intercalation. However, the O content is not reliable because we were measuring additional surface species which also contain oxygen. The Mo:Mg:O ratio were also quantified from the EDX spectra at the reacted region of the sample. They are included in the supporting information (Fig S17). They show Mg and O shuttling between charged and discharged states. The accuracy of S content from EDX is limited because of the S-K signal is fully overlapped with the Mo-L edge. We have also updated the main text accordingly.

As the reaction is progressing from the outside to the interior, the composition will continuously change so that a single number of bulk measurements by ICP would be fairly meaningless. Moreover, it is not possible to determine the O content by ICP because other sources of O (atmosphere, solvents used for digestion, dilution etc.) cause a large background in ICP analysis. Because of these concerns, we did not do the ICP of our electrodes.

2. In the EDX mapping, can the authors clarify the C elements also?

As the C-K signal is close to the limit of our EDX setup, we did not include it in the initial PCA analysis shown in the manuscript. However, because of the referee's comment, we included the C-K signal in the PCA analysis; and obtained a very good agreement with the PDF analysis further indicating the intercalation of solvated ions. The EDX C-K map and PCA analysis including C-K of the 30th discharged sample is included in the supporting information (Fig. S20). The principle components show the C, O and Mg are spatially correlated but uncorrelated to Mo and S.

3. What is the role of C in the MoS₂/C composite? Please provide a data that rationalize the exclusion of C as a host for charge storage. For example, what happens for MoS₂/C sample with increased composition of C? Or C itself as an electrode?

The dopamine acts as a regulation agent which helps to obtain MoS₂ nanosheets with smaller sizes. After sintering, the carbon species in MoS₂@C-PNR could improve the electric conductivity of the electrode material.

For answering to the questions relating to C, we performed further experiments. Pure MoS₂ nanomaterial (s-MoS₂) was synthesized through the same procedure without addition of dopamine. Meanwhile, the carbon species (s-C) were synthesized by hydrothermally treating

the dopamine/Tirs solution at 200 °C for 24 h followed by the same sintering procedure. We cycled both of them with the same condition. The results (Fig. S7) show that s-MoS₂ electrode exhibits similar behavior and offers comparable capacity (~100 mA h g⁻¹) as the MoS₂@C-PNR electrode during galvanostatic cycling, whereas the s-C electrode only delivers negligible capacity (<5 mA h g⁻¹) at this condition. By comparing Fig. S7b and Fig. 2c, it could be seen that s-MoS₂ electrode needs longer activation cycles which could be explained by the larger crystal size than the MoS₂@C-PNR electrode. These results indicate carbon contribution to the capacity is negligible. The manuscript has been revised to explain this.

4. What is the redox center that counter-balance the storage of solvated Mg²⁺, especially if the oxidation number of Mo remains the same during charge-discharge?

The phase transition from 2H to 1T is actually balancing the charge after solvated Mg²⁺ intercalation. The 2H-1T transition takes place only if there is electron doping to the 2H-MoS₂, which has been clarified by previous work (*ACS Nano* 8, 11, 11447-11453, *Adv. Energy Mater.* 2018, 8, 1703482, *MRS Bulletin*, 40(7), 585-591). The Mo4d orbitals in 2H-MoS₂ are split into three groups, namely dz², dxy dx²-y² and dyz dxz. There is ~1 eV energy gap between the first two band groups. The two Mo d electrons fill the lower dz² orbital which makes it semiconductive. For 1T-MoS₂, there are only two band groups (dxy dxz dyz and dz² dx²-y²) because of its octahedral coordination. The two Mo d electrons occupy the dxy dxz dyz orbitals but will not fill them. With the unfilled occupation, 1T-MoS₂ is conductive but not stable. But during solvated Mg²⁺ intercalation, the additional charges from cations will be transferred to MoS₂ and occupy the dxy dxz dyz orbitals which stabilize the 1T structure.

We appreciate the reviewer for this question because these descriptions are particularly important to readers with different backgrounds. This discussion has been included into the revised manuscript.

5. Any evidence from XANES, apart from XPS, which is too much surface-sensitive?

It is absolutely right that the XPS is only a surface sensitive technique. That is why we performed 4DSTEM measurements. The 4DSTEM based PDF analysis probes the material with high spatial resolution at different positions in the material. The method is a direct probe of the bonding information and sensitive to the modification of the atomic structure. XANES could only probe the electrode with micron-scale spatial resolution, while the TEM (4D-STEM) analysis shows that the reaction occurred in tens to a hundred nm from the surface. The majority of the XANES signal will come from the interior unreacted MoS₂ making the valence evaluation and chemical quantification more difficult.

6. The schematic figure 7 is problematic, because 1T phase persists according to XPS.

The XPS only probes the top surface (< 5nm) of the material. The adsorption of solvated Mg²⁺ or the Mg²⁺ residues at the surface, proved by XPS (Fig. S15) and EDX (Fig. S17b), stabilizes the 1T phase at the top surface of the material, even though the material was charged. To clarify

this, we performed local PDF analysis probing the structure variation of the charged material from the surface to the bulk. The results were added in the revised manuscript in Fig. 6d. This depth resolved local PDF analysis shows that the majority of the activated material are 2H phase, except for a thin layer (< 30nm) of 1T phase located at the surface. The back and forth switching between 2H and 1T during charging and discharging was evidenced by the averaged PDF data of the reacted area (from surface to ~100 nm deep) (Fig. 6a). Therefore, XPS showing 1T phase persists in the charged sample is a surface effect, thus does not represent the main reacted region of the material. We have adapted the main text of the manuscript to clarify it.

7. Equation 2 is also problematic. Is there any evidence for the removal of Mg²⁺ leaving DME solvents in the structure? For example, EDX of charged status shows simultaneous removal of O and Mg from the structure.

The EDX spectra (Fig. 4f) and corresponding quantification (Fig. S17b) show removal of most of the Mg and O by charging. It can be clearly seen that the Mg signal is approaching the detection limit in the charged sample, whereas still noticeable amounts of O can be detected. Thus the O to Mg ratio in the charged state is higher than that in the discharged state, suggesting some residual DME to be present.

Furthermore, the peak of C-O/C-C bonds of the charged PDFs (Fig. 6a and d) does not reduce down to the level as the PDFs of pristine sample (Fig. 6a, solid blue curve) and unreacted core (Fig. 6d, solid green curve), whereas the peak of Mg-O bonds of the charged PDFs almost disappeared comparing to that of the discharged PDF. This indicates an amount of C/O left in the material after the Mg²⁺ being extracted.

We added more detailed explanation in page 22 of the main text.

8. On the term “intercalation.” Is it really intercalation, given that it is surface reaction according to CV, without any reversible changes in the structure and oxidation number?

The reversible phase transformation between the crystalline 2H and strongly disordered (amorphous-like) 1T phase was observed by PDF and HRTEM. As discussed in the response to comment 4, the phase transformation is correlated with the charge transfer. The reversible crystallinity change strongly relates to the Mo-S bonding state, hence the oxidation state, such as the peak shift of the PDF (Fig. 6a) at 3.2 Å indicates a shorter Mo-Mo distance implying a reduction of Mo. Moreover, the STEM image (Fig. 4c) clearly shows the fragmentation of the material's structure with ~100 nm deep from the surface, which cannot be the result of a surface reaction only.

According to the literature (*Energy Environ. Sci.*, 2014, 7, 1597), in a potential independent capacitive system, there should be very little potential hysteresis between the charging and discharging steps particularly for slow charge-discharge times, which is not really present in our case. Therefore, the large capacitive contribution especially at high current rate is mainly contributed by the intercalation pseudocapacitance, which proves the fast kinetics of our cell configuration.

9. Can the authors provide similar experimental evidences on MoS₂-com, which can lead to clearer experimental observations due to the simplicity of the system, though the capacity is smaller?

We agree with the reviewer's concern about the carbon contribution to the total capacity. As discussed in comment 3, based on the reviewer's suggestion, we have performed new electrochemical cycling experiment on both pure MoS₂ (s-MoS₂) and pure carbon (s-C) samples (Fig. S7). The results indicate the carbon contribution to the capacity is negligible. In addition, the s-MoS₂ electrode exhibits similar activation process as the MoS₂@C-PNR sample in the first few cycles. Even the MoS₂-com electrode experiences the same activation process. Therefore, one could expect similar result to the carbon-free samples (s-MoS₂ or MoS₂-com). Actually, the idea to apply MoS₂@C-PNR as a model material instead of MoS₂-com is because the former could provide much higher capacity than the latter. The capacity corresponds to the amount of the intercalated solvated Mg ions as well as the intercalation reaction depth. Using MoS₂-com might increase the difficulty of experimental characterization and making the data interpretation more speculative, as the reaction would be limited to a very thin surface region. Moreover, from structure point of view, we clearly observed a reduced crystallinity, structure distortion, bond broken and short- and medium-range order evolution from discharging to charging cycles. All these occurred in the MoS₂ but not in the carbon.

10. On the claim of highly reversible deposition and stripping of Mg in Fig. S3. What is the coulombic efficiency from the CV? Is it close to 100%?

The coulombic efficiency (CE) of the electrolyte has been reported in our previous work (*J. Mater. Chem. A*, 2017,5, 10815-10820) by galvanostatic cycling, from which a CE>98% for 50 cycles has been shown. With higher purity of the electrolyte salt used in this work, this value is much improved which has been proved by the long-term Mg stripping/plating for more than 1000 h as reported in our recent publication (*ACS Energy Lett.* 3, 8, 2005-2013).

In this work, the highly reversible plating/stripping process is also evidenced by the tiny amount of net charge accumulation after a complete CV cycle as shown in Fig S4b of the revised version.

Overall, I do not feel that the paper will influence or facilitate innovative thinking in the field. Also, I feel that their claims of solvated intercalation require more strengthened experimental evidences. Some part of the manuscript may need proof-reading for clarity in terms of English grammar. On the other hand, I find positive sides of this work that suggests a general concept of utilizing a host material via co-intercalation of solvated Mg²⁺ in an Mg metal-compatible electrolyte. However, that still requires more convincing study on the charge storage mechanism. Because of these reasons, I recommend to postpone the publication of this manuscript until the authors provide more detailed examinations on the interaction between of solvated Mg²⁺ and MoS₂/C composite and more deeper understanding of the charge storage mechanisms via experimental or theoretical approaches.

With the oxygen incorporation seen by EDX in the discharged state and based on the PDF analysis where the Mg-O and C-C/C-O peaks in the PDF of the discharged sample (Fig.6a) and ICA analysis (Fig.6b) fits excellently with $\text{Mg}[\text{DME}]_x$, we have strong evidence for the incorporation from $\text{Mg}[\text{DME}]_x$ into the $\text{MoS}_2@\text{C-PNR}$ as discussed at the related parts of the manuscript. Furthermore, from the PCA analysis of the 30th discharged sample, the carbon peak always correlates to the Mg and O peak i.e. carbon follows Mg and O in the map concentrated on the surface shell. We believe that the present data is adequate to support our conclusion that solvated Mg^{2+} ions are intercalated into the MoS_2 structure. We appreciate if the reviewer could reevaluate the results presented in our revised manuscript.

Reviewer #3:

In the present manuscript, the authors successfully demonstrated reversible co-insertion / co-desertion of solvated Mg^{2+} ion into $\text{MoS}_2@\text{C-PNR}$. The authors conducted series of analytical study to understand the charge/discharge process of the $\text{MoS}_2@\text{C-PNR}$. Even though the concept of co-intercalation itself is not very new, the co-intercalation of organic solvent for Mg battery is very unique. Also the manuscript is well written. Therefore I think overall the present manuscript is deserved for publication in Nature Communications. However couple of the points below must be revised before publication.

1. Even with the thorough discussions concerning the charge/discharge mechanism, the proposed mechanism in Fig. 7 is still doubtful, because all the XRD patterns shown in Fig. S6 have peaks at 33° , 58° . If the MoS_2 layer is broken down into the small pieces during the discharging process: 2H-1T phase transition shown in Fig. 7, these peaks are supposed to disappear at the discharged state. At first the authors should reconfirm the peak assignment of these XRD peaks. Then, an alternate charge/discharge mechanism should be proposed. I think this point is very critical for this paper.

This comment helped us carefully to review the meaningfulness of the *ex situ* XRD results. In XRD experiments, we dominantly see the contribution of crystalline phases present in the material. If both crystalline and amorphous phases are present in a material, the amorphous phase is almost invisible as the coherent diffraction from the crystalline phase is so much stronger. As the MoS_2 is not transforming completely during charging and discharging, but the crystalline core of the $\text{MoS}_2@\text{C-PNR}$ still persists, this will dominate the X-ray diffraction pattern even when the outer parts of the rods completely turn into a disordered material. This is the reason why 4DSTEM and PDF are used to probe the disordered structure in the reacted region of the material.

2. The coulombic efficiency of the cell operated at 500 mA g⁻¹ is highest among all the cells operated at different current density. What is the potential side reaction during the slow charge/discharge process?

This is mainly related to the limited kinetics of the host during solvated Mg^{2+} de-/intercalation. The intercalated $\text{Mg}[\text{DME}]_x^{2+}$ cannot be extracted completely from the host during charging. Due to the strong electrostatic interaction, small amount of $\text{Mg}[\text{DME}]_x^{2+}$ will be stuck inside the host even at the fully charged state. From the XPS Mg spectra in Fig. S15, we could already see the Mg residues at the fully charged state are accumulating upon cycling. At low current density, the capacity is higher corresponding to more $\text{Mg}[\text{DME}]_x^{2+}$ intercalated into the MoS_2 layers, and they are able to go deeper inside. With the increase of the current density, the intercalated ions become fewer, and they are closer to the surface. In case of high current density, the diffusion pathway is relatively shorter so that fewer ions will be stuck, corresponding to a higher coulombic efficiency.

3. Discussions concerning the simulated PDF in Fig. S11 sounds strange to me. The author assigned the peak corresponding to (1st order) Mo-Mo and S-S at 3.2 Å shifts lower in the discharged state. However the Mo-S bond at 2.4 Å does not change. It shows shrinkage of the lattice constant along the a, b axis and expansion of the c-axis. It does not match with the XRD patterns in Fig. S6, which does not show any peak shift.

The nanocrystalline or fully disordered material we are analyzing by PDF is invisible in the XRD patterns, which are dominated by the residual (unreacted) crystalline MoS_2 at the core of the MoS_2/C rods as discussed in the response to comment 1. Moreover, the distances measured in PDF is a Fourier transformation of scattering power normalized diffraction which reveals direct atomic bonding distance but have no simple connection with any lattice distances directly seen in XRD or SAED. The peak positions do not correspond to any changes in the lattice constants.

As for the actual PDF analysis, the bond length of the covalent Mo-S bonds is very well defined and will not change significantly during charging (unless the type of bonding would change). This corresponds to the 2.4 Å seen in the PDF. The observed peak at ~3.2 Å corresponds to non-covalently bonded Mo to Mo and S to S distances as next nearest neighbor distances with the Mo-Mo distance dominating the peak as it is the much stronger scattered. While this peak is observed exactly at 3.2 Å in the pristine/charged state as expected for MoS_2 , the peak is slightly shifted in the discharged state indicating that the average Mo-Mo distance in the discharged state is slightly reduced. An explanation for the reduced Mo-Mo distance is breaking of some Mo-S bonds, which corresponds to a reduction of MoS_2 . This slight structural change will also affect the higher order Mo-Mo distances leading to a shift of the peak expected at 5.5 Å, so that it partially overlaps with the second order Mo-S distances at 5.1 Å appearing as one broad peak in the discharged state. We have updated the corresponding section in the manuscript to make this clearer.

4. The charge-discharge performance of the MoS_2/C -PNR at elevated temperature should be presented in the manuscript, because the authors concluded that the diffusivity of Mg^{2+} limits the capacity. Of course I understand that the concept of the present work is the novel cathode at ambient temperature, but the capacity at the elevated temperature provides good target for the future works.

According to this comment, new experiment of charge-discharge performance of the MoS₂@C-PNR at 45 °C was performed. The result is added in the revised manuscript. And it indeed delivers a significantly higher capacity (~160 mA h g⁻¹ at 50 mA g⁻¹ for 50 cycles) compared to the room temperature cycles as shown in Fig. S11.

REVIEWERS' COMMENTS:

Reviewer #1 (Remarks to the Author):

In the revised version of the manuscript the authors have positively answered to all the questions raised during my first revision. Thus, in my opinion, the manuscript should be accepted for publication in Nature Communications in this revised version.

Reviewer #2 (Remarks to the Author):

Overall, the authors revised the manuscript in accordance with my previous comments. Now I am convinced of the major claims of the manuscript. However, I still found some aspects that need to be correctly handled for the scientific accuracy as follow:

1. In p.10, onset potential is -0.4 V, not 0.4 V. The cyclic voltammogram in Fig. S4a evidences that actual efficiency is as low as ~80%. Comparison of the net charge is not fair because that includes the portion of charge by electrolyte oxidation at the higher potential.
2. P.11. ~0.2 V beyond the cutoff potential... : what does it mean?
3. P.22, Mo-S bond break and Mo reduction... : doesn't it mean that conversion reaction happened?
4. The authors did not supply ICP data because they thought the reaction happens in the surface only, having the interior part not reacted. Would the actual specific capacity significantly larger than what is reported in the manuscript if the reacted part is considered? Can the authors provide an expected value?

Reviewer #3 (Remarks to the Author):

The manuscript claims fast co-insertion/desertion of solvated Mg^{2+} into the $MoS_2@C$ -PNR. As I commented in the previous reviewing process, the concept is unique and manuscript is well written. Some unclear explanation concerning the XRD and PDF analyses were appropriately updated. Therefore, I think the updated manuscript is deserved for publication in Nature Communications.

I only have one suggestion for modification of Figure 7. It is better to draw bigger MoS_2 clusters in the Stage 3, because the current version reminds the reader that the co-insertion process initiates the complete structural collapse of the "crystalline" phase.

I would recommend the authors to redraw the Stage 3 using clusters with 9 of MoS_6 octahedral like Stage 2. I understand the authors already noted that the crystalline core remains as non-active part and only the amorphous phase works as active material, but many readers should misunderstand the reaction mechanism due to the completely decomposed shape of the MoS_2 clusters in Stage 3.

Again I would recommend that the revised version of the manuscript should be accepted with the minor update of the Figure 7.

Response to the reviewers' comments

We greatly appreciate the reviewers' constructive comments. In the final revision process, we took all the suggestions and comments and tried to deal with the remaining issues and concerns. The changes in the manuscript have been highlighted accordingly. Below are our point-by-point responses to the reviewers' comments.

Reviewer #1 (Remarks to the Author):

In the revised version of the manuscript the authors have positively answered to all the questions raised during my first revision. Thus, in my opinion, the manuscript should be accepted for publication in Nature Communications in this revised version.

We are grateful for the reviewer's comments.

Reviewer #2 (Remarks to the Author):

Overall, the authors revised the manuscript in accordance with my previous comments. Now I am convinced of the major claims of the manuscript. However, I still found some aspects that need to be correctly handled for the scientific accuracy as follow:

1. In P.10, onset potential is -0.4 V, not 0.4 V. The cyclic voltammogram in Fig. S4a evidences that actual efficiency is as low as ~80%. Comparison of the net charge is not fair because that includes the portion of charge by electrolyte oxidation at the higher potential.

We appreciate the reviewer's reminder. The onset potential value has been corrected in the final version of the manuscript.

In the CV measurement in Supplementary Fig. 3a (final version), we did not see obvious current response at high potential (2.0–4.5 V), which indicates a high oxidation stability. This result suggests neglectable contribution of electrolyte oxidation to the current (or charge). Therefore, the coulombic efficiency (CE) of the electrolyte is determined to be close to 100%, based on the accumulated charge after a complete CV cycle. If we understand correctly, the reviewer determined the CE value by simply using the current response ($24 \text{ mA} / 30 \text{ mA} = 80\%$). This is not correct. On the other hand, the CE value of the electrolyte has been reported in our previous study (*J. Mater. Chem. A*, 2017, 5, 10815-10820) by galvanostatic cycling, which shows a CE of >98%. In these senses, we still believe that our electrolyte has a high CE.

2. P.11. ~0.2 V beyond the cutoff potential... : what does it mean?

This sentence is talking about the CV curve of MoS₂@C-PNR cathode against the Mg counter electrode (Mg_{CE}) shown in Supplementary Fig. 4. In this curve, the potential range is around -0.2–2.7 V (vs. Mg_{CE}/Mg²⁺), which is ~0.2 V beyond the cut-off potential (0.01–2.5 V vs. Mg_{RE}/Mg²⁺ as shown in Fig. 2a). In order to get rid of this confusion, we rephrased this sentence in the final version.

3. P.22, *Mo-S bond break and Mo reduction... : doesn't it mean that conversion reaction happened?*

The Mo-S bond break and Mo reduction does not mean that conversion reaction happened. Both XPS peak fitting and PDF analysis of the 30th discharged sample indicate 2H-1T phase transition during cation intercalation. As discussed in P.16 of the final version, the 1T-MoS₂ is only stable with electron doping. With more electrons accommodating, the Mo valence of 1T-MoS₂ will decrease. But it still keeps the Mo-S bonds, just with different coordination. The phase transition is also accompanied by amorphization. During this process, some Mo-S bonds at the edges break with the reducing size of the MoS₂ nanosheet. Actually, there is no detectable Mo-Mo and Mg-S direct bond can be found in the PDF of discharged state, which strengthens the intercalation mechanism rather than conversion reaction. We have rephrased the sentence in the final version.

4. *The authors did not supply ICP data because they thought the reaction happens in the surface only, having the interior part not reacted. Would the actual specific capacity significantly larger than what is reported in the manuscript if the reacted part is considered? Can the authors provide an expected value?*

Yes, the specific capacity is larger if it is normalized by the mass of reacted zone only. However, since the size of the nanorods is not identical, it is difficult to determine the accurate value. Nevertheless, we were still trying to give a rough value. This requires estimation of the volume of activation shell. According to the TEM results, one could estimate it is in depth of ~100 nm. As shown in Fig. 1d, the diameter of the rods ranges from 500–800 nm and their length is several μm. Using a cylinder shape with 650 nm diameter and 5 μm length to approximate the MoS₂ rod, we can get the fraction of the activation shell is 54%. Namely, the actual capacity after 30 cycles at 10 mA g⁻¹ is determined to be 218 mA h g⁻¹.

In the final version of the manuscript, we still keep using the capacity value of 118 mA h g⁻¹, which is normalized by the total mass of the rods (including the unreacted zone). This is because we want to keep in line with other literatures, in which the specific capacity is usually normalized by the total mass of the active materials.

Reviewer #3 (Remarks to the Author):

The manuscript claims fast co-insertion/desertion of solvated Mg^{2+} into the $MoS_2@C-PNR$. As I commented in the previous reviewing process, the concept is unique and manuscript is well written. Some unclear explanation concerning the XRD and PDF analyses were appropriately updated. Therefore, I think the updated manuscript is deserved for publication in Nature Communications.

I only have one suggestion for modification of Figure 7. It is better to draw bigger MoS_2 clusters in the Stage 3, because the current version reminds the reader that the co-insertion process initiates the complete structural collapse of the "crystalline" phase.

I would recommend the authors to redraw the Stage 3 using clusters with 9 of MoS_6 octahedral like Stage 2. I understand the authors already noted that the crystalline core remains as non-active part and only the amorphous phase works as active material, but many readers should misunderstand the reaction mechanism due to the completely decomposed shape of the MoS_2 clusters in Stage 3.

Again I would recommend that the revised version of the manuscript should be accepted with the minor update of the Figure 7.

We are grateful for the reviewer's suggestion. In the final version of the manuscript, the MoS_2 clusters in the Stage 3 of Fig. 7 have been accordingly modified.